# DNA methylation-based high-resolution mapping of long-distance chromosomal interactions in nucleosome-depleted regions

Yi Li [1,2], James Lee[1,2] & Lu Bai [1,2,3] ✉

3C-based methods have significantly advanced our understanding of 3D genome organization. However, it remains a formidable task to precisely capture long-range chromosomal interactions between individual loci, such as those between promoters and distal enhancers. Here, we present Methyltransferase Targeting-based chromosome Architecture Capture (MTAC), a method that maps the contacts between a target site (viewpoint) and the rest of the genome in budding yeast with high resolution and sensitivity. MTAC detects hundreds of intra- and inter-chromosomal interactions within nucleosome-depleted regions (NDRs) that cannot be captured by 4C, Hi-C, or Micro-C. By applying MTAC to various viewpoints, we find that (1) most long-distance chromosomal interactions detected by MTAC reflect tethering by the nuclear pore complexes (NPCs), (2) genes co-regulated by methionine assemble into inter-chromosomal clusters near NPCs upon activation, (3) mediated by condensin, the mating locus forms a highly specific interaction with the recombination enhancer (RE) in a mating-type specific manner, and (4) correlation of MTAC signals among NDRs reveal spatial mixing and segregation of the genome. Overall, these results demonstrate MTAC as a powerful tool to resolve fine-scale long-distance chromosomal interactions and provide insights into the 3D genome organization.

The Chromosome Conformation Capture (3C) technique and its derivatives (4C, Hi-C, etc.) have revolutionized our understanding of the 3D genome organization[1–6]. By utilizing a crosslinking, digestion, and ligation process, these methods have revealed extensive interactions that occur between linearly distant chromatin fragments. Nevertheless, 3C-based methods come with certain limitations. The resolution of 3C is ultimately constrained by DNA fragment sizes after digestion. In addition, due to the stochastic nature of chromosomal interactions, 4C or Hi-C signals often need to be averaged over large genomic intervals to improve the signal-to-noise ratio, which further reduces resolution to $10^3$–$10^5$ base pairs (bps)[7]. Consequently, these measurements are best at revealing large-scale structures over

hundreds to thousands of kilobases (kbs), including Rabl configuration, A/B compartments, and topologically associated domains (TADs)[8–11]. However, some biologically relevant chromosome conformations, like enhancer-promoter (E-P) or promoter-promoter (P-P) interactions, involve highly dynamic contacts between short regulatory regions that typically span a few hundred bps. Mapping these types of interactions with 3C-based methods are challenging, even with a large number of cells and high sequencing depth[12–14].

A variant of the Hi-C technique, Micro-C, uses smaller restriction fragments generated by Micrococcal Nuclease (MNase) digestion and thus provides higher resolution[15,16]. Micro-C was shown to be more effective in detecting interactions between regulatory regions,

[1]Department of Biochemistry and Molecular Biology, The Pennsylvania State University, University Park, PA 16802, USA. [2]Center for Eukaryotic Gene Regulation, The Pennsylvania State University, University Park, PA 16802, USA. [3]Department of Physics, The Pennsylvania State University, University Park, PA 16802, USA. ✉e-mail: lub15@psu.edu

including E-P or P-P interactions[17,18]. It should be noted that many E-P or P-P chromosomal contacts may be mediated by proteins associated with nucleosome-depleted-regions (NDRs). The interactions between NDRs cannot be directly measured by Micro-C because these open regions are digested away by MNase. Therefore, the measured E-P or P-P contacts likely rely on the crosslinked products in the neighboring nucleosomal regions, which prevents a precise mapping of the interaction site. Moreover, differential sensitivity to MNase digestion among genomic regions may also introduce bias in the measured contact frequencies.

Since crosslinking, digestion, and ligation potentially generate biases, alternative methods free of these reaction steps have been explored. Some of these methods, including TSA-seq and DamC, target ectopic DNA modifying enzymes to specific genomic loci and use the resulting modifications to identify loci that are physically proximal to the target sites[19,20]. However, these methods also suffer from low resolutions. The resolution of DamC, for example, is ultimately limited by the frequency of the Dam motif, "GATC", which on average occurs once a few hundred bps. Dam's preference for accessible regions further reduces the number of usable motifs, necessitating the data to be binned over larger distances. As a result, similar to Hi-C, these methods have been mostly used to study large 3D genome structures like lamina/nuclear speckle associated domains[21-24] or TADs[20].

Here, we develop a <u>M</u>ethyltransferase <u>T</u>argeting-based chromosome <u>A</u>rchitecture <u>C</u>apture (MTAC) method to measure finer-scale long-distance chromosomal interactions. This method utilizes an ectopic DNA methyltransferase, M.CviPI, that selectively methylates the cytosine in a "GC" dinucleotide[25]. M.CviPI fused to LacI can be recruited to an integrated LacO array and methylate accessible "GC"s that are physically proximal in *cis* and in *trans*. Like 4C, MTAC allows us to probe the interaction between one locus (viewpoint, VP) and the rest of the genome (one-to-all). However, in comparison with DamC and 4C, this method offers two major advantages. First, it has higher resolution. Over 98% of yeast NDR sequences contain "GC"s (in contrast to 34% containing GATC). Most of these NDRs have more than one "GC", allowing us to integrate the MTAC signals over multiple methylation sites to reduce measurement noise. This enhanced signal-to-noise ratio allows us to pinpoint single NDRs that are in contact with the target site. Second, unlike 3C-based methods that take a "snapshot" of the genomic interactions at a single time point, MTAC captures interactions by leaving stable methylation marks on DNA, allowing the interactions to be recorded cumulatively over a period of time. This way, MTAC provides a much higher sensitivity to long-distance interactions that tend to be highly dynamic[26,27].

We test the utility of MTAC by inserting the LacO array near nuclear pore complex (NPC) associated regions, methionine-regulated (*MET*) genes, and the mating locus (*MAT*). Using a NDR-centric analysis, we detect remarkably specific interactions between the target sites and individual NDRs near other NPC-proximal loci, co-regulated *MET* genes, and recombination enhancer (RE) / telomeres, respectively. Importantly, most of these interactions cannot be detected by 4C, Hi-C, or Micro-C. By manipulating VPs and/or applying other genetic perturbations, our measurements further suggest that (1) NPC tethering is a major mechanism that mediates long-distance chromosomal interactions in budding yeast, (2) at least a subset of the *MET* genes clusters near NPCs upon activation, and the clustered *MET* genes have closer proximity to each other than to other NPC-associated genes, (3) The *MAT*-RE interaction is mediated by condensin in a mating-type specific manner, and (4) The NPC-associated regions are well mixed, while *MAT* / RE / telomere together occupies a separate nuclear space from the bulk of the genome. Overall, these results demonstrate that MTAC is a powerful method to map fine-scale chromosomal interactions and probe 3D genome organization.

## Results

### MTAC captures long-distance chromosomal interactions with high resolution and sensitivity

The MTAC method is based on methylation of DNA physically proximal to a target site. A GpC methyltransferase, M.CviPI, is fused with LacI and recruited to the VP where a LacO array was inserted. The recruited M.CviPI methylates nearby cytosines in a "GC" context both in *cis* and in *trans*, which can be detected by methylated DNA immunoprecipitation (MeDIP) sequencing[28] (Fig. 1A). The methylation pattern of this yeast strain ("targeted strain") is compared with a control strain with no LacO insertion, which reflects the background methylation by free LacI-M.CviPI. The loci that interact with the VP are defined as regions that have significantly higher methylation levels in the targeted strain than the control strain (Fig. 1A).

To ensure a high signal-to-noise ratio in the MTAC measurement, the fusion protein needs to have a high affinity towards LacO sites and a low expression level to limit non-specific methylation. Based on our previous study of LacI binding[29], we fused M.CviPI to the C-terminus of LacI (LacI-M.CviPI) and used the LexA-VP16-ER induction system to drive its expression[30]. We tuned the LacI-M.CviPI level by varying the concentration of β-estradiol and found that 2 nM β-estradiol generates the largest fold difference in local DNA methylation between targeted and control strains (Supplementary Fig. 1A, B). We also tested different induction times (1 or 2 h) and LacO repeat numbers (4X, 8X, 16X, and 256X). Among these parameters, longer induction time and more LacO repeats yield more significant MTAC signals (Supplementary Fig. 1C). All the data shown below are therefore generated with 2 nM β-estradiol induction for 2 h with a 256X LacO array.

We constructed an MTAC strain with the VP on chromosome VII, next to the *YBP2* gene (*YBP2* VP) (Fig. 1B). This VP was chosen because it is located in a gene-rich region near many NDRs that would provide robust local MTAC signals. We performed Hi-C experiments in this strain ± LacO insertion and ± LacI-M.CviPI induction (Materials and Methods). No significant changes in the Hi-C signals are detected, indicating that the LacO array and DNA methylation have no major impact on the nascent genome organization (Supplementary Fig. 2). After applying the MTAC procedure in Fig. 1A to this strain, most methylation is detected within NDRs, which are more accessible to methyltransferase[31] (Supplementary Fig. 3A). In comparison to the control, we observed hypermethylation near the VP in the targeted strain (Fig. 1B), with the fold enrichment decreasing exponentially with distance to VP (Fig. 1C). These short-range *cis* interactions with the VP likely reflect the local folding of the chromosome. Similar local patterns are found among different VPs that we tested (Supplementary Fig. 3B, C).

MTAC also detects long-distance *cis* and *trans* interactions. When the data are binned with a constant interval, the high methylation signals from NDRs tend to be "diluted" by the low signals from adjacent nucleosomal regions, leading to reduced enrichment. To solve this problem, we took an NDR-centric approach by integrating the methylation signals over each NDR throughout the genome and calculated the statistical difference of the integrated signals between targeted and control strains (Fig. 1A) (Materials and Methods). Using log2 (Fold Change) > 0.7 and adjusted *P*-value < 0.05 as cutoffs (same cutoffs for all MTAC data in this study), we identified 22 intra-chromosomal interactions with *YBP2* VP that are over 30 kb in distance ("far-*cis*") and 89 inter-chromosomal interactions ("*trans*") (Fig. 1D), with a few examples shown in Fig. 1E. The averaged MTAC signals and heatmaps of interacting vs non-interacting NDRs are shown in Supplementary Fig. 3D. Notably, the interactions detected by this method are highly localized: in Fig. 1E "*trans* (2)", for example, significantly enriched signal was detected in the right-most NDR, but not the neighboring NDRs 1, 3, or 7 kb away, even though some of these NDRs show comparable basal-level methylation. When all the interacting NDRs are aligned by their centers, the average MTAC

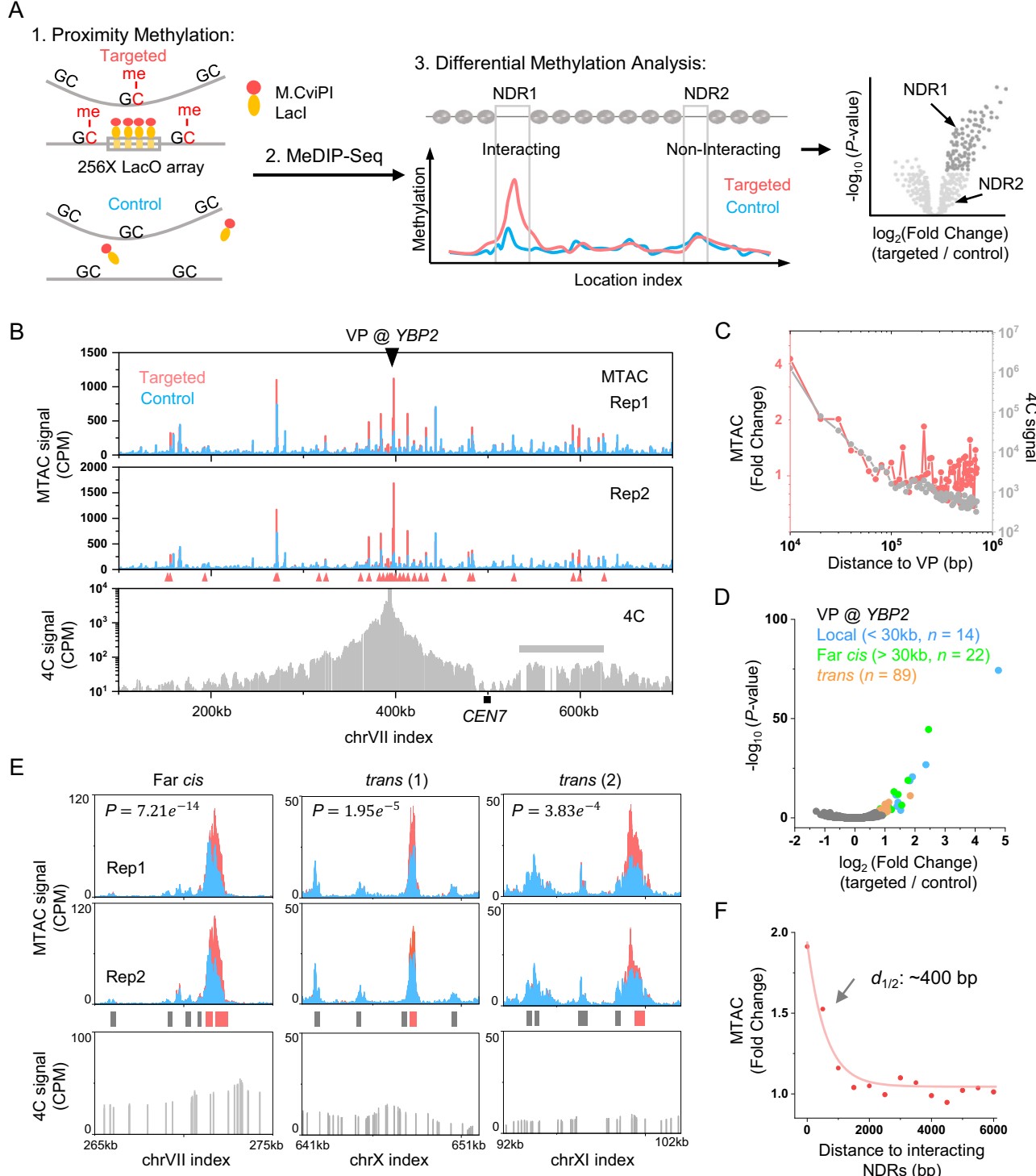

**Fig. 1 | MTAC captures long-distance chromosomal interactions with high resolution and sensitivity. A** Schematic diagram of MTAC methodology. 1. Constructs of the targeted and control strains, as well as the expected proximity methylation. 2. MeDIP is used for detecting methylated cytosines. 3. Differential methylation analysis for individual NDRs in targeted vs control strains. Fold change and *P*-values for each NDR between these two strains are shown in a volcano plot. **B** MTAC data at the *YBP2* viewpoint (VP). A 256-repeat LacO array was inserted into a gene-rich region near the *YBP2* gene. Methylation data near the VP are shown for the targeted (pink) vs control strain (blue) (*n* = 2 biologically independent samples per group). The 4C signals with the same VP in the same region are shown in the lower panel (gray). The interacting regions called by MTAC or 4C are shown in red triangles and gray bars, respectively. CPM: counts per million (same as below).

CEN7: Chromosome VII centromere. **C** Averaged MTAC (pink) and 4C signals (gray) as a function of distance to the *YBP2* VP. **D** Volcano plot of MTAC signals from the *YBP2* VP. The local, far-*cis*, and *trans* NDRs that pass the threshold are shown in blue, green, and orange, respectively. **E** MTAC data at three regions that make long-distance interactions with the *YBP2* VP. The 4C signals over the same regions are shown in the bottom panels. Non-interacting NDRs and interacting NDRs are shown as gray and red bars, respectively. **F** Averaged long-distance MTAC signals as a function of distance to the centers of the captured interacting NDRs. Statistical tests for (**A**), (**D**), and (**E**) are based on two-sided Wald test with *P*-value adjustment for multiple comparisons (Benjamini & Hochberg method) by DESeq2 (*n* = 2 biologically independent samples per group).

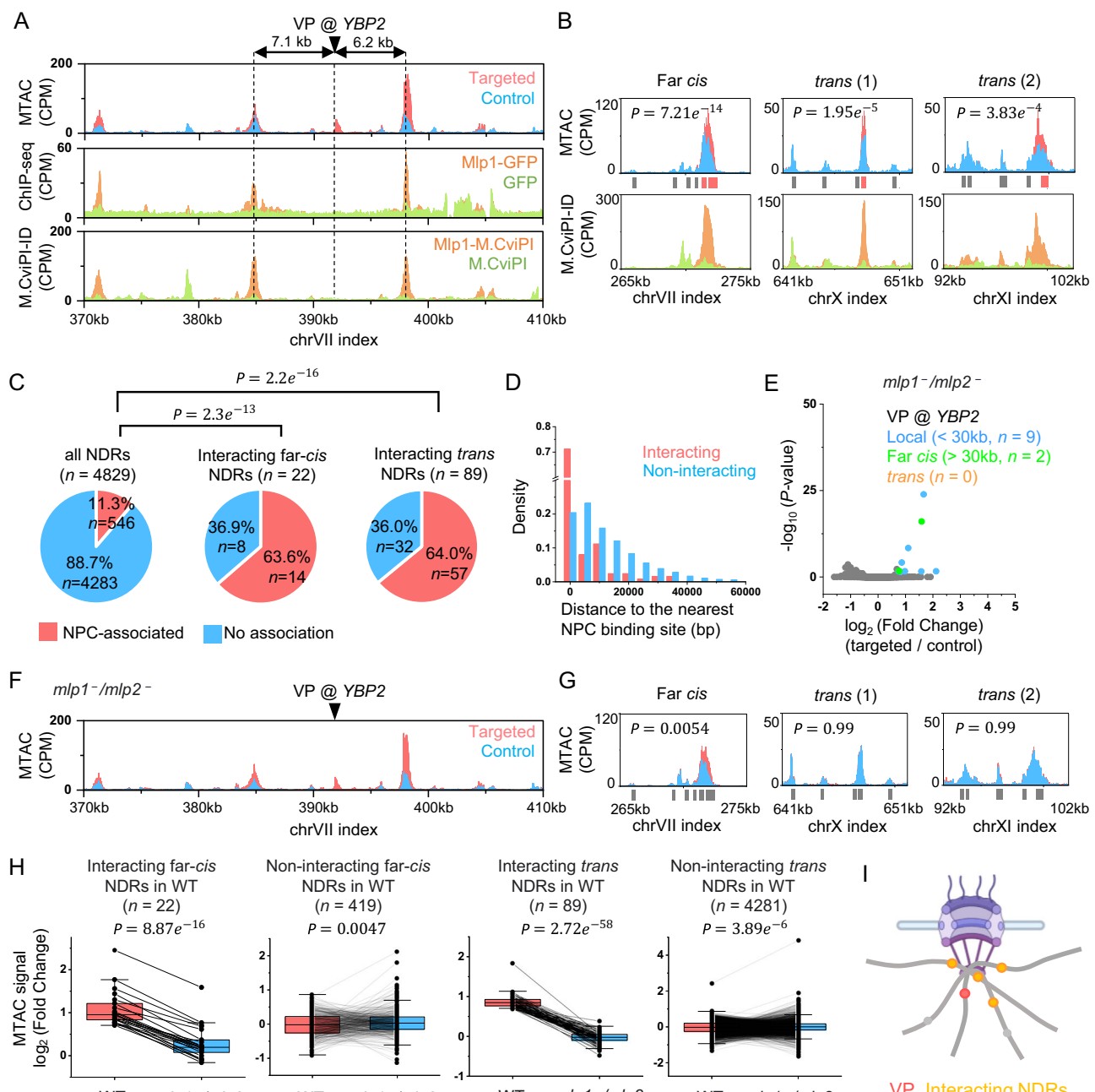

**Fig. 2 | MTAC captures nuclear pore complex (NPC)-associated interactions from the *YBP2* VP. A** MTAC, Mlp1 ChIP-seq[33], and Mlp1 M.CviPI-ID data from the *YBP2* VP. For ChIP-seq data, ChIP-seq signal against Mlp1-GFP and free GFP (control) are shown in orange and green. For M.CviPI-ID data, methylation signals from M.CviPI-tagged Mlp1 and free M. CviPI (control) are shown in orange and green. **B** MTAC and Mlp1 M.CviPI-ID data at three regions that make long-distance interactions with the *YBP2* VP. **C** Percentage of NPC-association by genome-wide NDRs vs NDRs that make long-distance interactions with the *YBP2* VP. **D** Histogram of distances between interacting / non-interacting NDRs and the nearest NPC binding site. **E** Volcano plot of MTAC signals from the *YBP2* VP in the *MLP1/MLP2* double-deletion strain. **F** MTAC data near the *YBP2* VP (local interactions) in the *MLP1/MLP2* double-deletion strain. **G** MTAC data at the same regions as in B. With *MLP1/MLP2* deletion, these regions no longer make significant long-distance interactions with

*YBP2* (the fold change of the far-*cis* interaction does not meet our cutoff). **H** Comparison of MTAC signals between WT and *MLP1/MLP2* mutant over interacting and non-interacting regions. Box plot notation: center line, median; box limits, 25th and 75th percentiles; boundaries of the whiskers, minimum and maximum values; points and lines, pairwise comparison of actual datapoints in two groups. **I** Model of NPC-associated interactions. The VP (red) tethered by NPC falls in proximity with other NPC-associated regions (orange). Created with Biorender.com. Statistical tests for (**B**), (**E**), and (**G**) are based on two-sided Wald test with *P*-value adjustment for multiple comparisons (Benjamini & Hochberg method) by DESeq2 (*n* = 2 biologically independent samples per group). Statistical test for (**C**) is based on two-sided two proportions z-test. Statistical test for (**H**) is based on two-sided paired students' *t*-test. Source data are provided as a Source Data file.

signals in the neighboring NDRs decrease exponentially with a half-distance of ~400 bp (Fig. 1F). Given that the average distance between neighboring NDRs in the yeast genome is 2–3 kb, this data shows that MTAC measurement can pinpoint the long-distance

interaction site to 1–2 NDRs, which reflects the high resolution of this method.

We next compared the interactions measured by MTAC vs 3C-based methods. Since the scale of MTAC is analogous to 4C, we

performed 4C with the *YBP2* VP using the standard protocol (Materials and Methods). Consistent with other published 4C data, most signals originate from local *cis* interactions, and the only long-range interaction that can be identified is a ~90 kb intra-chromosomal region from chrVII: 535 kb to 624 kb, which overlaps with a few interacting NDRs detected by MTAC (Fig. 1B). This region has a similar distance to the chrVII centromere as the VP, suggesting that the elevated contacts may be attributed to the Rabl configuration[8]. None of the other long-distance interactions (including far-*cis* and *trans* interactions) detected by MTAC are identified as significant 4C peaks (examples shown in Fig. 1E), although the compiled 4C signals over the MTAC far-*cis* or *trans* interacting NDRs are mildly but significantly higher than the non-interacting regions (Supplementary Fig. 3E). The local contact frequencies measured by 4C follow the same trend as the MTAC signals up to ~20 kb, but MTAC captures more interactions at longer distances (Fig. 1C). These observations illustrate the higher sensitivity of MTAC to long-distance interactions.

For further comparison, we binned our Hi-C data in yeast with 100 M reads into 10 kb windows. No long-distance interaction with the *YBP2* locus can be called from the Hi-C data, except for those caused by the Rabl configuration. When we collected the Hi-C signals from all the long-distance interacting regions detected by MTAC, the Hi-C signals from the far-*cis* regions, but not the *trans* ones, showed slight but significant elevation (Supplementary Fig. 3F). Additionally, we compared MTAC to a published Micro-C-XL dataset in yeast with 500 M reads[16]. As Micro-C requires genome digestion by MNase, which eliminates most NDRs, the interacting regions detected by MTAC are generally missing from the Micro-C data at 100 bp resolution. Especially for *trans* interactions, which tend to be sparser, we need to bin the Micro-C data at 5 kb to get enough reads for statistical comparison. At this resolution, a slight increase of Micro-C signal was observed between the *YBP2* VP and far-*cis* interacting NDRs, but not for the *trans* interaction sites (Supplementary Fig. 3G). Again, no individual contact point can be identified. Finally, we selected 25 interactions detected by MTAC with various signal intensities, as well as five negative control regions, and tested them by 3C (Supplementary Fig. 4A). 21 out of 25 MTAC-detected interactions can be detected by 3C, while 4 out of the 5 negative regions do not show 3C signals (Supplementary Fig. 4B). Taken together, these results show some consistency between MTAC and 3C-based methods, but more importantly, they demonstrate the unique power of MTAC in identifying highly localized, point-to-point interactions.

## MTAC captures nuclear pore complex (NPC)-associated interactions from the *YBP2* VP

Out of 4829 genome-wide NDRs, MTAC shows that *YBP2* specifically interacts with 22 far-*cis* and 89 *trans* NDRs (Fig. 1D). What is special about these NDRs, and how do they interact with the *YBP2* locus? We speculated that these NDRs may bind to certain factors that mediate the interactions. We therefore compared the MTAC signals with previously published Chromatin Immuno-precipitation sequencing (ChIP-seq) and Chromatin Immuno-precipitation with Exonuclease (ChIP-exo) data[32,33] to search for factors that are associated with these *YBP2*-interacting NDRs. Interestingly, the binding sites of a nuclear basket protein, Mlp1, match closely to *YBP2*-interacting regions (Fig. 2A). This finding suggests that *YBP2* interacts with other NPC-associated genes in *cis* and in *trans*.

To further test this idea, we conducted our own measurement of Mlp1 binding (Materials and Methods). To be more compatible with MTAC, instead of using ChIP-seq, we used a modified version of DamID called M.CviPI-ID to methylate Mlp1-associated sites using M.CviPI-labeled Mlp1 (Supplementary Fig. 5A, B). The methylation data was then compared with a control strain containing free M.CviPI to identify enriched sites (Materials and Methods). The M.CviPI-ID agrees well with the published Mlp1 ChIP-seq data, albeit with a better signal-to-

noise ratio (Supplementary Fig. 5C). Importantly, MTAC peaks largely overlap with Mlp1-M.CviPI-ID peaks, both for the local and long-distance interactions (Fig. 2A, B). To evaluate this more quantitatively, we aligned M.CviPI-ID peaks with genome-wide NDR coordinates, resulting in 546 NPC-associated NDRs (11.3% of the total NDRs). Strikingly, out of the 22 and 89 NDRs that have far-*cis* and *trans* interactions with *YBP2*, 14 (63.6%) and 57 (64.0%) are associated with NPCs, much higher than the genome-wide average (Fig. 2C). Even for the ~1/3 of the interacting NDRs that are not directly NPC-associated, they tend to be NPC-proximal (median distance 6.9 kb vs 10.6 kb genome-wide; *P*-value = 6.3e-3) (Fig. 2D).

The results above strongly indicate the critical role of NPC in mediating long-distance genomic interactions with *YBP2* VP. To further test this idea, we deleted *MLP1* and its paralog *MLP2*, as some studies have suggested that these two proteins have largely redundant functions[34,35]. We performed MTAC on the same *YBP2* VP. In this *MLP1/MLP2* double-knockout mutant, the local interactions of *YBP2* remain largely intact (Fig. 2E, F), but most of the far-*cis* interactions and all the *trans* interactions are lost (Fig. 2E, G, Supplementary Fig. 6A, B). Quantitively, MTAC signals over long-distance interacting NDRs are significantly decreased, but the ones over non-interacting NDRs are slightly increased in the mutant (Fig. 2H, Supplementary Fig. 6A), indicating a global deregulation of genome organization. Together, these results show that the contacts with *YBP2* VP from distant genomic loci are likely due to the tethering by NPCs, and Mlp1 / Mlp2 are essential to mediate these interactions (Fig. 2I).

## VP dynamics facilitate the capture of NPC-associated interactions

The *YBP2* locus is not immediately adjacent to NPC (7.1 and 6.2 kb away from the upstream and downstream Mlp1 binding peaks, respectively) (Fig. 2A). This raises questions about whether NPC-mediated interactions can be detected from any genomic location, and how MTAC signals are influenced by the distance between VP and NPC. More specifically, we suspected that MTAC may detect more and/or stronger interactions when the VP is closer to NPC-associated loci. We thus generated two more strains with the LacO array inserted near *YBP2* at *PUS2* and *OLE1*, which are 1.5 and 2.3 kb away from the nearest Mlp1 binding site, respectively (Fig. 3A). Surprisingly, MTAC only captured 30 and 8 long-distance interactions from the *PUS2* VP and the *OLE1* VP, respectively, much less than the 111 interactions detected at *YBP2* (Fig. 3B). Most of these *PUS2* and *OLE1* interacting loci are a subset of those found at *YBP2*, and again, they tend to be NPC-associated (Fig. 3C, D). These results suggest that all three VPs capture NPC-mediated interactions, but the one located slightly away from NPCs can capture these interactions more effectively.

To test the generality of the findings above, we applied MTAC to seven other VPs on three different chromosomes, with distances to NPCs ranging from 0.7 kb to 49 kb. It is worth noting that this range covers most of the genomic loci, as the average distance to the closest NPC from a random genomic locus is ~12 kb. All VPs tested here were found to interact with NPC-associated NDRs (Fig. 3D). The number of local interactions is relatively stable at different VP-to-NPC distances, while the numbers of far-*cis* and *trans* interactions increase sharply when the VP is ~6 kb away from the NPC (Fig. 3E). As VPs move further away from the NPCs, some of their interacting NDRs also become more distant to NPCs (Fig. 3F). These data support the model in Fig. 3G, where VP dynamics facilitate the capture of NPC-associated interactions. When the VP is very close to the NPC contact point, its motion becomes more constrained, limiting the number of NDRs it can encounter. In contrast, more distant VPs can explore a larger volume and contact more genomic loci, including those that are further away from the NPCs.

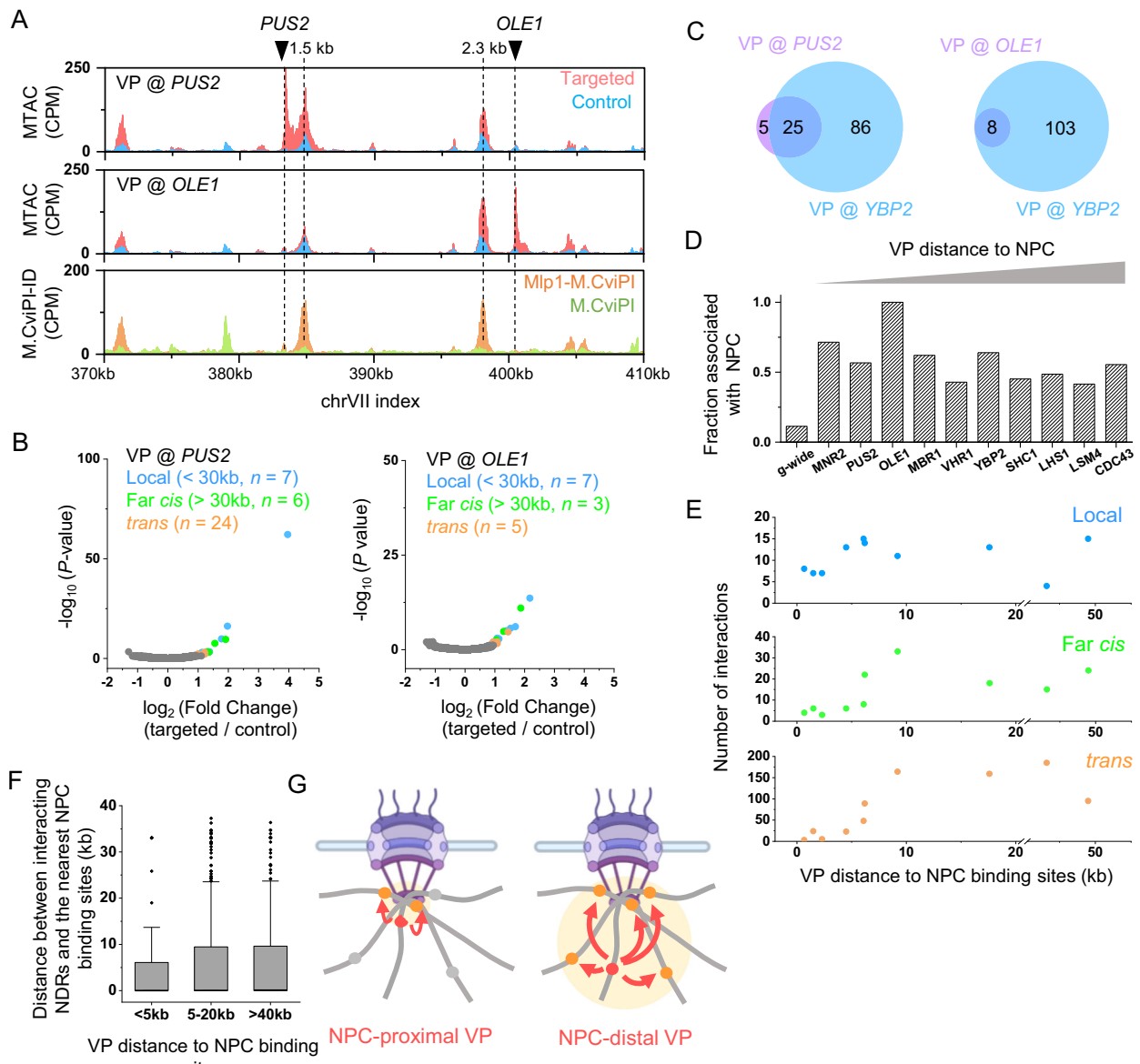

**Fig. 3 | VP dynamics facilitate the capture of NPC-associated interactions.**
**A** MTAC and M.CviPI-ID data from the *PUS2* and *OLE1* VPs, which are 1.5 and 2.3 kb away from the nearest Mlp1 binding sites. **B** Volcano plots of MTAC signals from the *PUS2* and *OLE1* VPs. **C** Venn diagram of the long-distance interacting loci of *PUS2* / *OLE1* VPs and *YBP2*. **D** Fraction of interacting loci that are associated with NPC at different VPs. The VPs are ordered based on their distance to the nearest NPC binding sites. g-wide: genome-wide (same as below). **E** Number of local, far-*cis*, and *trans* interactions as a function of distance between VPs and the nearest NPC binding sites. **F** Distance between interacting loci and the nearest NPC binding sites captured by VPs with different distances to NPC binding sites. Less than 5 kb: *n* = 74

NDRs, 5-20 kb: *n* = 545 NDRs, > 20 kb: *n* = 319 NDRs. Box plot *n*otation: center line, median; box limits, 25th and 75th percentiles; boundaries of the whiskers, minimum and maximum values; points, outliers. **G** Model explaining the dependence of MTAC signals on NPC-to-VP distance. NPC-distal VPs can capture more interactions due to the increase of chromosome motion. Created with Biorender.com. Statistical test for (**B**) is based on two-sided Wald test with *P*-value adjustment for multiple comparisons (Benjamini & Hochberg method) by DESeq2 (*n* = 2 biologically independent samples per group). Statistical test for (**F**) is based on two-sided students' *t*-test. Source data are provided as a Source Data file.

## MTAC detects clustering among activated *MET* genes

Having demonstrated MTAC as a powerful method for capturing NPC-mediated interactions, we next investigated if it could detect chromosomal interactions between co-regulated genes. Our previous 3C measurements indicated that a methionine regulated gene, *MET13*, contacts a few co-regulated genes, and such interactions showed higher strength with methionine depletion (activating condition; -met)[36]. To further probe the proximity among the *MET* genes, we carried out Hi-C in -met conditions to measure the contact frequencies among genome-wide Met4 targets (Materials and Methods) (Supplementary Fig. 7). We also carried out the same analysis using published

Hi-C maps in +met[37] (Supplementary Fig. 7). In both conditions, these Hi-C datasets fail to detect any significant interactions between individual pairs of Met4-targeted sites. Even the compiled contact maps for all pairs do not show significant signals (Fig. 4A). Since Met4 targets are distributed across many chromosomes, these data are consistent with Supplementary Fig. 3E that it is hard to detect inter-chromosomal contacts from Hi-C[14]. We thus applied MTAC to measure the proximity among *MET* genes in ±met conditions. We first performed MTAC by placing the VP at *TRP2*, which is 3 kb away from a Met4-targeted gene, *MET6* (Fig. 4B). In the +met repressive condition, only five local and one far-*cis* interactions were detected, whereas in -met, five *trans*

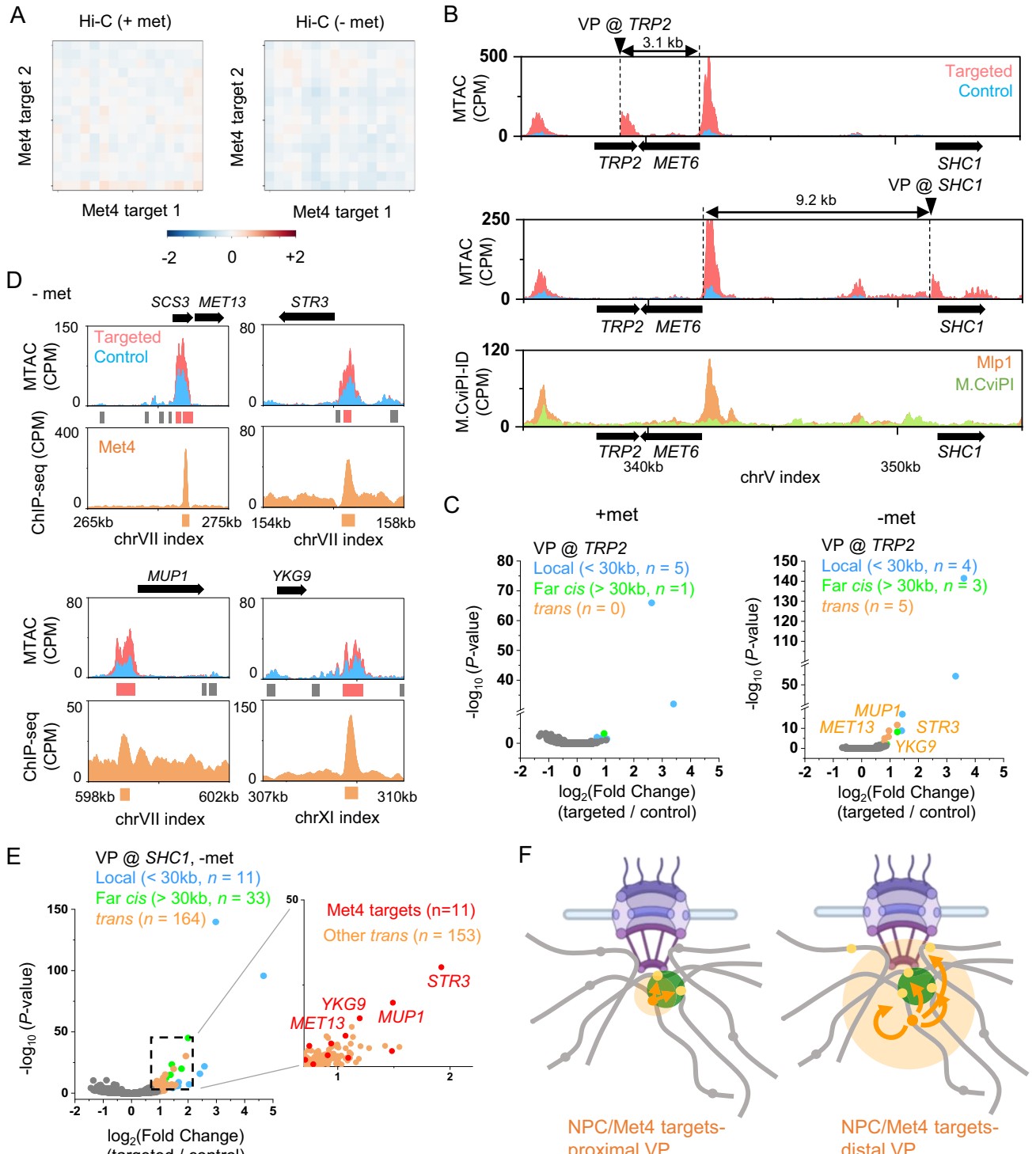

**Fig. 4 | MTAC captures the clustering among activated MET genes. A** Hi-C aggregate plot on Met4-regulated genes in −met (our data) and +met (Bastié et al.[37].) conditions. The Hi-C signals (log2(observed/expected)) are binned with a 10 kb resolution. Signals centered around Met4 ChIP peaks are aggregated and plotted within +− 70 kb regions. **B** MTAC and Mlp1 M.CviPI-ID data near the *TRP2* and *SHC1* VPs. The *TRP2* and *SHC1* VPs are 3.1 kb and 9.2 kb from a Met4-regulated gene, *MET6*. **C** Volcano plot of MTAC signals from the *TRP2* VP in ± met conditions. **D** MTAC and Met4 ChIP-seq data over the five *trans*-interacting NDRs at the *TRP2*

VP. Two of these NDRs are adjacent to the *MET13* gene. Met4 ChIP-seq peaks are shown as orange bars. **E** Volcano plot of MTAC signals from the *SHC1* VP in −met condition. Inset is a zoom-in volcano plot of the *trans*-Interacting NDRs. Met4 targets are shown in red and other *trans*-interacting NDRs are shown in orange. **F** Clustering of the activated *MET* genes detected by proximal vs distal VPs. Created with Biorender.com. Statistical tests for (**C**) and (**E**) are based on two-sided Wald test with *P*-value adjustment for multiple comparisons (Benjamini & Hochberg method) by DESeq2 (*n* = 2 biologically independent samples per group).

interactions emerged (Fig. 4C). Strikingly, all five *trans*-interacting NDRs are proximal to Met4 binding sites (two of them are adjacent to *MET13*) (Fig. 4D). This overlap is highly significant (*P*-value < 1e-5), especially given that Met4 ChIP-seq only shows 45 peaks throughout the entire genome.

The number of interactions captured by MTAC in Fig. 4C is small, potentially because *MET6* is NPC-associated, and the VP used above is very close to the NPC contact point (Fig. 4B). We therefore moved the VP to *SHC1*, a locus ~9 kb away from *MET6* (Fig. 4B), and performed MTAC in -met. From this more distal VP, MTAC indeed detected many more long-distance interactions (Fig. 4E). Consistent with our observations in Fig. 2 and 3, many of these interactions (89 out of 197) are NPC-associated. Notably, the most significant *trans*-interactions with the *SHC1* VP occur near the same Met4 targets detected from the *TRP2* VP (Fig. 4C and E). These observations have interesting implications on the spatial organization of the *MET* regulon (see discussion) (Fig. 4F).

## MTAC reveals the mating type-specific interactions between the mating locus and RE

Long-distance chromosomal interactions have been proposed to play a significant role in a well-studied phenomenon in budding yeast called the mating type switch. The two mating types of budding yeast, a and α, are controlled by sequences in the mating locus (*MAT*) on chromosome III (chrIII). During each mother cell cycle, *HO* endonuclease is expressed and generates a double strand break (DSB) in *MAT*, which is repaired by a homologous sequence, *HMLα* or *HMRa*, on the same chromosome (Fig. 5A). To ensure effective mating type switching, *MATa* cells prefer to use *HMLα* as a donor for homologous repair, while *MATα* cells prefer *HMRa*. Such "donor preference" is promoted by a locus near *HMLα* called the Recombination Enhancer (RE) (Fig. 5A). Many genetic studies have indicated that *MAT* and RE physically interact in *MATa* cells, which leads to the preferred usage of *HMLα*[38–41]. However, previous 3C-based measurements, including our own 4C data (see below), did not detect localized contacts between *MAT* and RE, and only showed mild differences of chrIII conformation in a vs α cells[38].

To test if MTAC can better pinpoint the interactions with *MAT*, we inserted the LacO array next to the *MAT* locus and performed the MTAC in both mating types (Fig. 5A). In *MATα* cells, only local interactions were detected (Fig. 5B). In contrast, MTAC captured 4 far-*cis* and 19 *trans* interactions in *MATa* cells, 13 of which are at telomeric or sub-telomeric regions (<10 kb from the chromosome ends) (Fig. 5B). Remarkably, one of the far-*cis* interaction sites is directly over RE (Fig. 5C). We also detected interactions with *BAR1*, a gene that is only expressed in *MATa* cells (Fig. 5C), although the functional significance of this interaction is unclear. No significant interactions between *MATa/α* and *HML/HMR* were found (Fig. 5C), which may be due to intrinsically low contact frequencies and/or the lack of extensive NDRs in *HML/HMR*. None of these specific interactions are detected by 4C. In particular, 4C data shows elevated contact frequencies over the left arm of chrIII in *MATa* cells, but such elevation is diffusive without distinct peaks (Fig. 5C).

To further corroborate our findings, we forced the cells to switch mating type by expressing the *HO* endonuclease controlled by the *GAL1* promoter[42,43]. Following the procedure in Supplementary Fig. 8A, we generated a pulse of HO expression that led to the mating-type switch in ~47% of the cells (Supplementary Fig. 8B & C). As a control, the cells continuously incubated with glucose show no switching (Supplementary Fig. 8B, C). We applied MTAC using the same *MAT* VP to the population after switching, which contains a mixture of *MATa* and *MATα* cells. The interaction patterns are consistent with a mixed population: cells that were originally *MATa* lose most of their long-distance contacts, while those originally *MATα* gain interactions with a few telomeres (Supplementary Fig. 8D, E). These findings confirm the

interactions with *MAT* found in Fig. 5B and illustrate their dynamic changes during the mating type switching process.

Previous studies indicate that RE, Sir2, and condensin are important for *MATa* interactions[42,44]. To test if these elements indeed lead to changes of MTAC signals, we constructed strains containing a RE or *SIR2* gene deletion, or auxin-degradable Smc2 (Smc2-AID) (Materials and Methods). Since Smc2-AID shows a partial degradation (reduced to ~50% of wt level) in the presence of ubiquitin ligase OsTIR1 even without auxin (Supplementary Fig. 8F), we compared Smc2-AID strains with or without OsTIR1. We applied MTAC to these cells using the same *MATa* VP and measured its interaction to RE and a telomeric region (chrXII: 5422-5571) by MeDIP-qPCR. Consistent with Fig. 5B, *MATa* in the wt cells shows interactions with RE and telomere, as indicated by the significant differences in their MeDIP signals between targeted and control strains (Fig. 5D). These interactions disappear in the strains with RE or *SIR2* gene deletion (Fig. 5D). In the condensin-depletion strains, these interactions can be detected with Smc2-AID in the absence of OsTIR1 (Smc2-AID only) but become weaker or non-significant in the presence of OsTIR1 with or without auxin (Fig. 5E), which shows that even a partial condensin loss can compromise these long-distance interactions. These results support the critical roles of RE, Sir2, and condensin in establishing *MATa* interactions. They also demonstrate that MTAC can be used to probe the mechanisms of long-distance chromosomal interactions through mutational perturbation.

## MTAC reveals a partitioned occupancy of nuclear space by different genomic loci

In comparison to other VPs tested in this study, the *MATa* VP has a unique interaction pattern. All the other VPs tend to make interactions with NPC-associated NDRs (Fig. 3D), and therefore, their contacts have significant overlaps (Supplementary Fig. 9). In contrast, the contacts made by *MATa* VP are not enriched with NPC association and have almost no overlap with the other VPs (Fig. 5F and Supplementary Fig. 9). This observation implies that *MATa* is physically isolated from the bulk of the genome, presumably through interaction with the telomeres. It also raises an interesting possibility that the correlation of the MTAC data can reveal co-localization or separation in nuclear space among genomic loci. More specifically, NDRs located in overlapped nuclear space should make similar interactions with VPs, and therefore, their MTAC signals should be positively correlated, while NDRs in separate nuclear space should show the opposite trend.

To test this idea, we collected the MTAC signals from all interacting NDRs (N = 487) with different VPs and calculated the pair-wise Pearson's correlations among these NDRs (Fig. 6A). We then clustered the correlation coefficients hierarchically (Materials and Methods) and plotted using a heatmap (Fig. 6B). Six major clusters emerged with strong positive correlations within each cluster and weaker or even negative correlations in-between (Fig. 6B). The most striking negative correlation occurs between cluster 6 and all the other clusters (Fig. 6B). Consistent with the observations in Supplementary Fig. 9, the vast majority of the NDRs in cluster 6 are the ones that interact with the *MATa* VP (35 out of 39), including *MAT*-proximal regions, RE, telomeres, and subtelomeric regions. A telomeric region (chrXV:15-149) that was previously not captured by *MATa* VP also appears in cluster 6. In comparison with other clusters, cluster 6 is the least associated with NPC (Fig. 6B). These data again suggest that *MATa*, RE and telomeres occupy a distinct nuclear space and separate from other NPC-associated regions.

For Cluster 2–5, the majority of NDRs in each cluster belong to a single chromosome that contains VPs tested in this study (Fig. 6C). Most of the interactions made by these NDRs are thus local and far *cis* interactions with the VPs on the same chromosomes (Fig. 6D). In contrast, the NDRs in Cluster 1 are more evenly distributed on different chromosomes, making mostly *trans* and far *cis* interactions with the VPs (Fig. 6C, D). Therefore, these clusters also reflect spatial

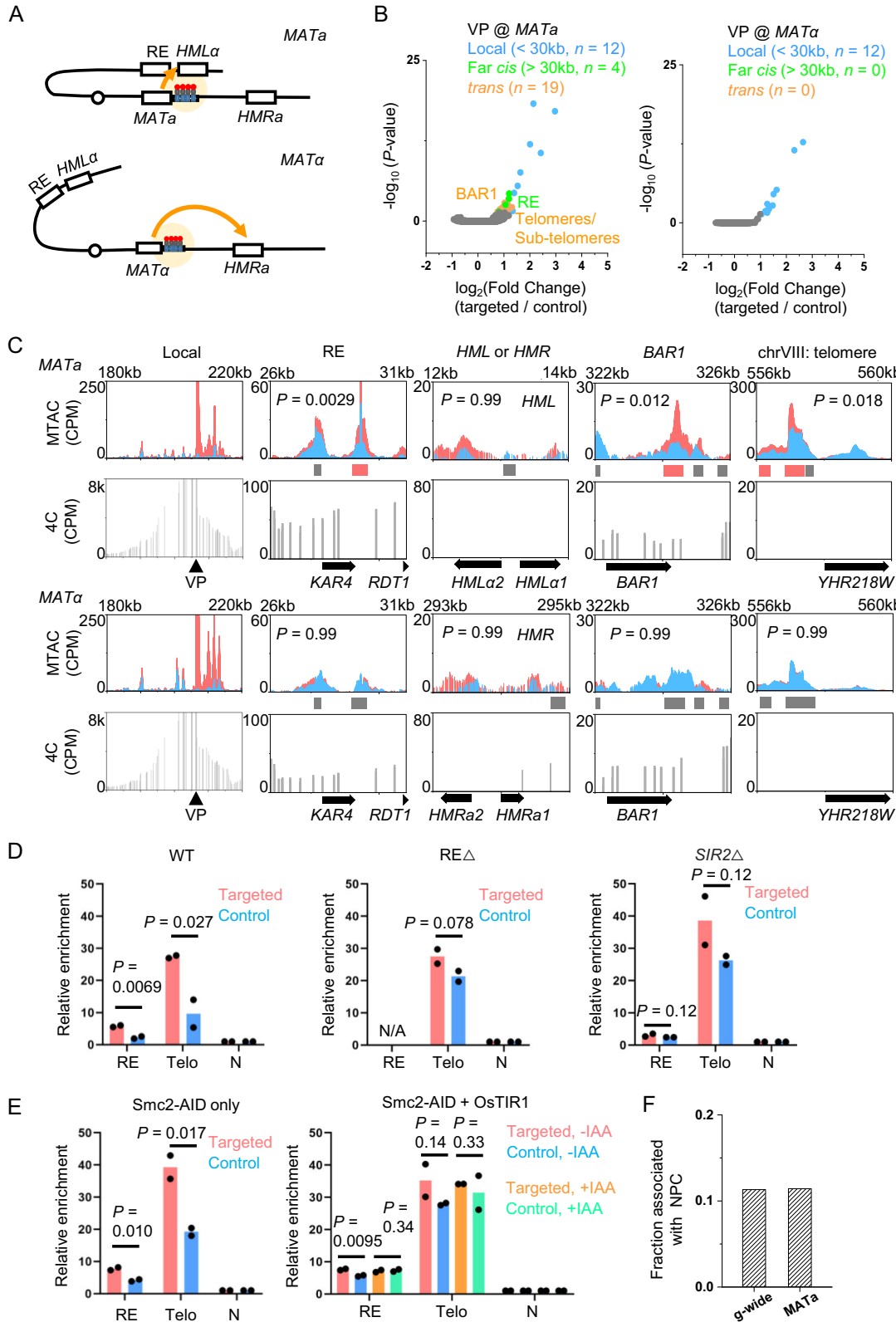

**Fig. 5 | MTAC reveals the mating type-specific interactions of the mating locus.**
**A** Scheme of MTAC with the *MATa* (top) or *MATα* (bottom) VP. **B** Volcano plot of
MTAC signals from the *MAT* VP in *MATa* and *MATα* cell. *MATa* specifically interacts
with RE, *BAR1*, telomeres and sub-telomeric regions. **C** Representative tracks of
MTAC and 4C data near the *MAT* VP and other captured interacting regions in both
mating types. Elevated methylation signals are observed in RE, *BAR1*, and telomere
of chrVIII only in *MATa* cell. **D**, **E** MeDIP-qPCR of interacting RE and a telomeric
region of *MATa* viewpoint in WT, ΔRE, *sir2⁻*, and Smc2 depleted strains (n = 2

biologically independent samples per group). **F** Fraction of the interacting NDRs at
the *MATa* VP that are associated with NPC. Statistical tests for (**B**) and (**C**) are based
on two-sided Wald test with *P*-value adjustment for multiple comparisons (Benja-
mini & Hochberg method) by DESeq2 (n = 2 biologically independent samples per
group). Statistical tests for (**D**) and (**E**) are based on one-sided students' *t*-test (n = 2
biologically independent samples per group). Source data are provided as a Source
Data file.

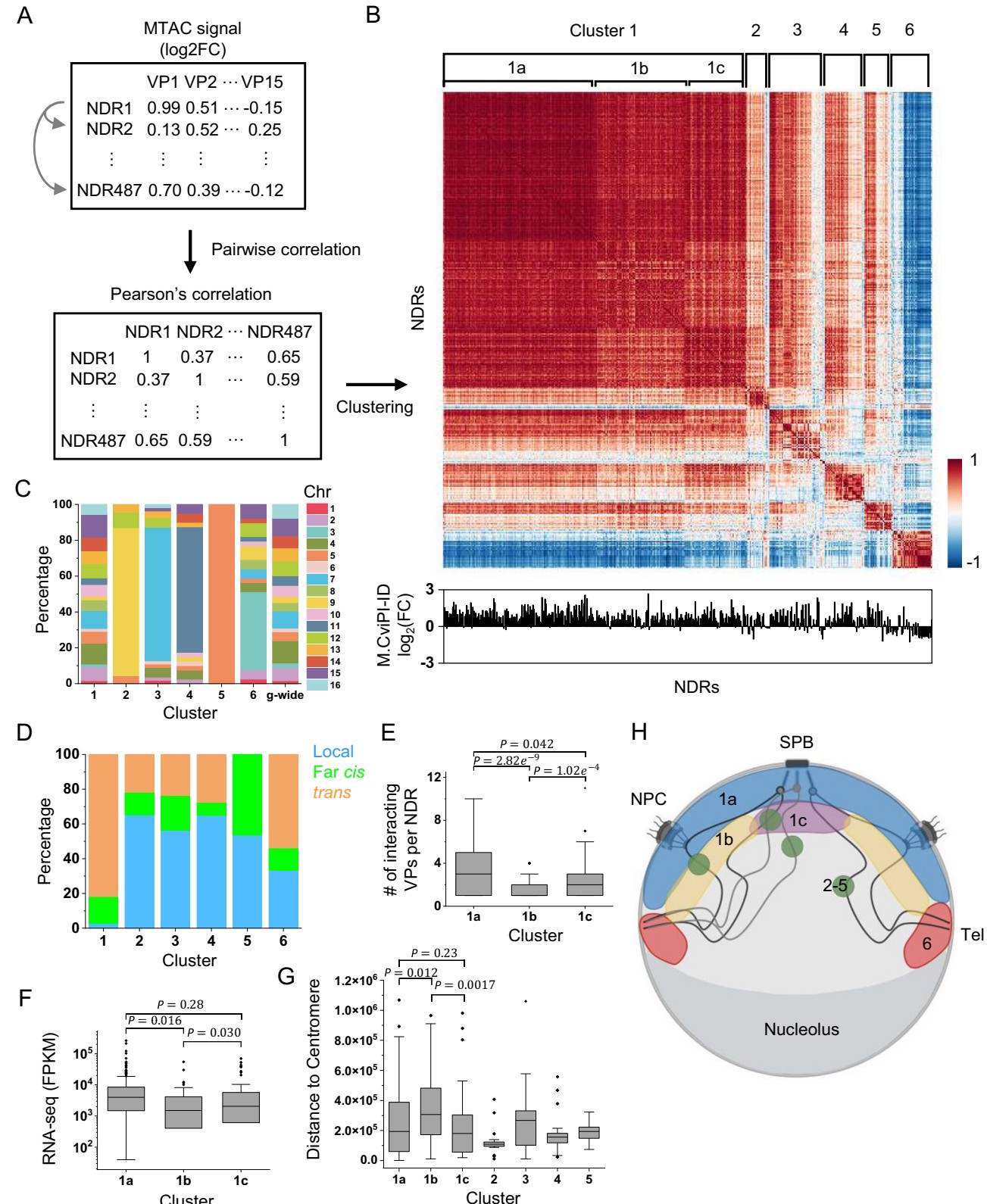

separation: Cluster 1 makes long-distance chromosomal interactions in the proximity of NPCs, while many NDRs in Cluster 2−5 contact VPs due to local folding, which can occur anywhere in the nuclei.

Cluster 1 can be further divided into sub-clusters a-c, where the correlations are higher within portion 1a and weaker in 1b and 1c. NDRs in cluster 1a have the strongest association with NPCs (Fig. 6B) and interact with significantly more VPs than those in 1b and 1c (Fig. 6E).

Further analysis of the RNA-seq data shows that genes adjacent to cluster 1a NDRs tend to have large RNA-seq counts (Fig. 6F). Overall, these data indicate that cluster 1a represents highly expressed genes that are more stably associated with NPCs. When comparing cluster 1b and 1c, we noticed that 1c contains regions closer to the centromere (Fig. 6G). Consistent with this observation, 1c also shows higher correlations with centromere-proximal clusters 2 and 4, while 1b is more

**Fig. 6 | MTAC reveals the partitioned usage of nuclear space. A** Scheme of correlation clustering analysis. MTAC signals from 15 VPs are calculated for 488 NDRs that are captured to be interacting regions with at least one of the VPs. Pairwise Pearson's correlation is calculated for each NDR based on their MTAC signals from each VP. Correlation coefficient matrix is then clustered hierarchically to generate the heatmap in B. Cluster 1: $n = 302$ NDRs, cluster 2: $n = 23$ NDRs, cluster 3: $n = 55$ NDRs, cluster 4: $n = 40$ NDRs, cluster 5: $n = 28$ NDRs, cluster 6: $n = 39$ NDRs. **B** Clustered correlation heatmap. Six distinct clusters are found. Cluster 1 can be further divided into 1a-c. Bottom: Mlp1 M.CviPI-ID signals for the corresponding NDRs. **C** Percentage of chromosomes on which NDRs locate in each cluster. Genome-wide NDR locations are plotted as comparison. **D** Interaction types (local, far *cis*, and *trans*) of NDRs in each cluster. If multiple types of interactions are made by the same NDR, local interaction is counted before far *cis*, and far *cis* before *trans*.

**E** Number of interacting VPs per NDR in cluster 1a, 1b and 1c. Cluster 1a: $n = 154$ NDRs, cluster 1b: $n = 85$ NDRs, cluster 1c: $n = 63$ NDRs. **F** RNA-seq counts of nearest gene of NDRs in cluster 1a ($n = 154$ genes), 1b ($n = 82$ genes), and 1c ($n = 61$ genes). FPKM, Fragments Per Kilobase of transcript per Million mapped reads. ($n = 2$ biologically independent samples per group) **G** Distance to centromere of NDRs in each cluster. Cluster 1a: $n = 154$ NDRs, cluster 1b: $n = 85$ NDRs, cluster 1c: $n = 63$ NDRs, cluster 2: $n = 23$ NDRs, cluster 3: $n = 55$ NDRs, cluster 4: $n = 40$ NDRs, cluster 5: $n = 28$ NDRs. **H** Model of the nuclear distributions of NDRs in different clusters. Created with Biorender.com. Box plot notation (**E**), (**F**), and (**G**): center line, median; box limits, 25th and 75th percentiles; boundaries of the whiskers, minimum and maximum values; points, outliers. Statistical tests for (**E**), (**F**), and (**G**) are based on two-sided students' *t*-test. Source data are provided as a Source Data file.

---

correlated with centromere-distal clusters 3 and 5 (Fig. 6B and G). Taken together, the correlations of MTAC signals with multiple VPs among NDRs reveal a few major structural components in the nuclei that allow the partitioning of the nuclear space, including NPC, telomere, and centromere (Fig. 6H).

## Discussion

In this work, we developed MTAC, a proximity methylation assay for capturing genome-wide chromosomal interactions with a single VP. In comparison with 4C, a widely-used method to measure genomic interactions at the same scale, MTAC offers some unique advantages. It is crosslinking, digestion, and ligation free, and therefore avoids biases generated in these steps. The procedure is simple, involving less reactions than typical 4C, Hi-C or micro-C. The use of a methyltransferase recognizing "GC" dinucleotides provides a higher resolution than 3C-based methods, as well as methods based on Dam methylase, such as DamC. The methylation marker is largely stable, allowing signals to accumulate over time and thus making it more likely to capture weaker and more dynamic interactions. Integrations of the MTAC signals over genome-wide NDRs, most of which contain multiple "GC"s, reduce the noise of the methylation measurements and enhance signal-to-noise ratios. Indeed, this work clearly demonstrates that MTAC can detect many long-distance interactions that are not captured by 4C. These advantages are accompanied by some drawbacks as well. The MTAC method requires genomic editing with long LacO repeats, which may be challenging for some species and/or cell types. The insertion points of LacO need to be carefully chosen to avoid perturbation of cellular function. Also, it is hard for MTAC to be scaled up to detect all-to-all interactions like Hi-C.

Besides establishing the method, this work also generated insights on the principles of 3D organization of the budding yeast genome. First, our results suggest that most interactions detected by MTAC are mediated through NPC tethering. Importantly, this does not mean that the bulk of the genome is constantly associated with NPCs. In contrast, NPC binding is likely to be a dynamic process[45], but chromosomal encounters in the untethered state may be too transient to be captured by MTAC. A recent study in mammalian cells shows that NPC associated regions are enriched in super-enhancers and active genes, and are hubs for chromatin structural proteins, including CTCF and PolII[46]. Therefore, tethering by NPC could be one of the conserved mechanisms in eukaryotes that generates more stable long-distance chromosomal interactions for regulations of active transcription. Moving the VP closer to the NPC decreases the number of captured interactions, which is consistent with previous reports that tethering to the NPC reduces the chromatin sub-diffusion[45]. More generally, by manipulating VP positions, MTAC can function as a "ruler" to detect genomic loci that have variable distances to a tethering point.

Chromosomal interactions may also manifest as clustering of co-regulated genes. Recent studies in budding yeast have presented evidence that a few regulons tend to cluster upon activation, including genes responding to galactose or heat shock[47-49]. Most of these

interactions were detected through imaging, which does not have the resolution to pinpoint the interactions between specific regulatory regions. Here, using a VP near *MET6*, a gene induced by methionine depletion, MTAC detects remarkably specific interactions among Met4-regulated genes only under the activating condition. Interestingly, an adjacent VP to *MET6* only captures interactions with co-regulated genes, while a more distal VP shows additional interactions with NPC-associated regions, indicating that the *MET* genes in the cluster have closer proximity to each other than to other NPC-associated genes. These results also suggest that the mechanism of *MET* clustering is different from NPC tethering. Based on previous findings that the clustering of *GAL* and heat shock genes rely on their corresponding transcription activators[47,49,50], we speculate that Met4, or some other co-activators in the methionine-response pathway, may mediate *MET* gene interactions. The detailed mechanism requires further elucidation.

ChrIII architecture has been studied extensively due to its connection to mating type switch and donor preference[38]. Despite genetic evidence that RE makes specific interactions with the *MAT* locus, 3C-based methods only found diffusive interactions between *HML* and the right arm of chrIII, and such interactions are only mildly enhanced in the MATa cells[38]. The proximity between *MAT* and RE was also probed by imaging[38,51], but the co-localization measurement was limited by the optical resolution. MTAC, on the other hand, captures a point-to-point interaction between *MAT* and RE in a-type cells. Out of the 44 NDRs on the left arm of chrIII, the NDR over RE and two NDRs on telomere are the only ones showing significant MTAC signals, indicating that the interaction between *MAT*a and RE is highly specific. In terms of mechanism, published ChIP-seq data shows strong binding of condensin on the right side of RE near the ARS304 and *RDT1* gene[42], and deletion of this region abolishes the mating type-dependent Hi-C pattern of ChrIII[38]. Consistent with this observation, we found that the depletion of condensin subunit Smc2 compromises the MTAC signal between *MAT*a and RE. Functionally, condensin depletion impairs mating-type switch efficiency and donor preference[42,44]. Together, this data provides strong support for the model that condensin mediates the looping between *MAT*a and RE to promote the usage of *HML* as the sequence donor.

All analyses above examine which NDRs are contacted by each VP. By collecting interaction data from multiple VPs, we can also ask the reciprocal question, i.e. which VPs are contacted by each NDR. For NDRs with a very similar interaction pattern, it is likely that they occupy similar nuclear space. Through this analysis, a few major spatial patterns start to emerge. First, NPC-associated NDRs across all 16 chromosomes (cluster 1) show high and relatively uniform correlations to each other. Except for a small degree of separation based on the proximity to centromere (cluster 1b vs 1c), there are no significant sub-clustering among these NDRs, indicating that these NPC-associated genes are well mixed without major NPC specialization (Fig. 6H). Second, NDRs that engage in local interactions with the VPs (cluster 2–5), especially those that do not bind to NPCs, show low correlations

with NPC-associated NDRs. This reflects different mechanisms of interaction (local folding instead of NPC tethering) and also different locations of interaction (whole nuclei instead of on the membrane) (Fig. 6H). Finally, the NDRs close to the *MAT* and telomeres / subtelomeric regions (cluster 6) show striking anti-correlations with NPC-associated NDRs, which indicates a spatial separation between NPC and telomeres, even though they are both anchored to the nuclear envelope (Fig. 6H). These observations are consistent with previous imaging studies showing the clustering of telomeres[52,53]. They also suggest that telomeres are attached to the envelope through different complexes other than NPCs, like the nuclear membrane proteins Mps3 and / or Esc1[54,55]. It should be noted that the correlation data in Fig. 6B depends on the VPs used in this study, and other patterns may emerge when more VPs are explored. Nevertheless, with the current VPs, MTAC is shown to be a powerful method in probing long-distance chromosomal interactions and 3D genome organization.

MTAC should have broad applications in other genetically tractable species, including mammals, with some key considerations: (1) GC content in NDRs. Ideally, multiple GCs should be present in individual NDRs to ensure a high signal-to-noise ratio. This criterion should be met in human cells, where NDRs are typically couple of hundred bps in length and GC dinucleotide occurs once every ~20 bp[56,57]. (2) Stability of the GC methylation. Unlike yeast, many species undergo an active 5mC demethylation process. Although TET enzymes in human cells have a strong preference for 5mCG, they have lower activities towards 5mC in other sequence context[58]. It thus needs to be carefully evaluated if G5mC is stable enough for MTAC to maintain the interaction information. (3) Native DNA Methylation. Endogenous 5mC may interfere with MTAC by elevating the background signal in the control and reducing the signal-to-noise ratio. Switching from meDIP to bisulfite sequencing may be beneficial in this case to allow the differentiation of 5mC from different sequence context[59].

## Methods

### Plasmid and strain construction
Standard methods were used to construct the strains and plasmids (See Supplementary Table 1 for plasmids and strains information). All strains used in this study were derived from a w303 background. MTAC control strain was constructed by integrating a *LacI-NLS-M.CviPI* into the *ADH1* locus. This fusion gene is driven by an engineered core *GAL1* promoter that binds to LexA-ER-VP16, which can be transcriptionally induced by β-estradiol[30,60]. MTAC targeted strains were constructed by integrating a plasmid containing 256X LacO repeats derived from a previous study[43] into the corresponding VP locus into the control strain.

For strains used in Mlp1 M.CviPI-ID experiments, Mlp1 M.CviPI-ID strain and control were constructed by integrating a *Mlp1-M.CviPI-GFP* or *NLS-M.CviPI-GFP* driven by the same β-estradiol induced promoter into the *ADH1* locus. For strains used in the mating-type switching assay, a plasmid containing *GAL10pr-HO* was integrated into the *GAL10* promoter locus to express HO in both the MTAC targeted and control strains with the *MAT* VP. For strains with auxin-induced degradation, we tagged the C-terminus of endogenous Smc2 with AID and a V5 tag, and introduced OsTIR1 to the cells[61]. Depletion of the Smc2 was achieved by adding auxin (Sigma, I2886) to a final concentration of 500 μM. Strains were incubated with auxin for 1 h prior to MTAC experiment. Depletion was confirmed by western blot.

### MTAC induction and DNA preparation
Yeast strains were grown in 50 mL synthetic media with or without methionine at 30 °C to OD660 = 0.3. β-estradiol (Sigma, 50-28-2) was then added to a final concentration of 2 nM for 2 h, allowing the *LacI-NLS-M.CviPI* fusion protein to express, bind the LacO array, and methylate the neighboring DNA. To extract genomic DNA, cells were harvested, resuspended in 1 mL spheroplasting solution (1 M sorbitol, 0.5 mM 2-mercaptoethanol, 0.5 mg/mL zymolyase), and incubated at 37 °C for 30 min. After a brief centrifugation, cell pellets were resuspended in lysis buffer (65 mM EDTA, 52 mM Tris-HCl, 0.5% SDS; pH 8.0) and incubated at 65 °C for 30 min. Cell lysate was then sonicated with Diagenode Bioruptor Pico to shear DNA (EZ mode, 30 s on, 30 s off, 10 cycles). Genomic DNA was purified by standard phenol-chloroform DNA extraction. A total of 5–10 μg genomic DNA was expected. Gel electrophoresis was performed to confirm the shearing efficiency. Samples with DNA fragment size ranging from 300 bp to 600 bp were used for downstream immunoprecipitation and sequencing library preparation.

### MeDIP-seq
Immunoprecipitation of methylated DNA was adapted from a previously published protocol[62]. 5 μg of DNA were diluted in 400 μL TE buffer, mixed with 100 μL 5X IP buffer (0.7 M NaCl, 50 mM sodium phosphate, 0.25% Triton X-100; pH 7.0) and 2 μg 5-Methylcytosine Monoclonal Antibody (Thermo Fisher, 33D3), and incubated at 4 °C overnight with rotation. After the overnight incubation, 80 μL of protein A/G agarose beads (Santa Cruz, sc-2003) were added to the DNA-antibody mixture. The mixture was then incubated at 4 °C for 2 h with rotation. Beads were washed for three times with 1 X IP buffer and resuspended in 210 μL digestion buffer (50 mM Tris-HCl, 10 mM EDTA, 0.5% SDS; pH 8.0). 20 μL of 20 mg/mL proteinase K (Thermo Fisher, EO0491) was added to the digestion solution and incubated at 55 °C for 2 h. Reaction was filtered with Pierce spin-filtering column (Thermo Fisher, PI69700). DNA was precipitated with ethanol and resuspended in TE buffer. qPCR was performed to check the enrichment of regions near the VP in both the MTAC targeted and control strain before library preparation (Supplementary Table 2). MeDIP-seq libraries were prepared with NEBNext Ultra II DNA Library Prep Kit (NEB, E7645L). Next generation sequencing was performed on NextSeq 2000 (Illumina). Two to five million 50 bp paired-end reads were expected for each MTAC sample. For detecting interactions on repetitive or homologous regions, 150 bp reads can be used instead.

### MTAC data analysis
Paired-end reads of 50 bp or 150 bp were processed with fastp (version 0.23.2) to remove low quality reads. Processed reads were aligned to *S. cerevisiae* genome (sacCer3) with BWA (version 0.7.17). Aligned bam files were deduplicated with Picard MarkDuplicates (version 3.0.0). These steps typically remove ~30% of the reads. For the remaining reads, bigwig files were generated by deeptools bamCoverage (version 3.5.1) for visualization purposes. To detect differentially methylated NDRs, aligned reads on each NDR were counted by Subread feature-Counts (version 2.0.3) with the parameters "-p -O". The NDR coordinates were generated by applying a previously described NDR annotation model[63] to MNase-seq data in Kubik et al. (Supplementary Data 1)[64]. Differential methylation analysis was performed on data from the targeted and control strains. Fold change and *P*-value were generated by DESeq2 (version 1.38.0) without outliers filtering. log2 (Fold Change) > 0.7 and the adjusted *P*-value < 0.05 were used as the cutoff for identifying differentially methylated regions.

### MTAC correlation analysis
For the correlation analysis in Fig. 6, we collected MTAC data for 487 interacting NDRs from 15 VPs located on 5 different chromosomes. In other words, each NDR has 15 MTAC signals (log2FC) associated with different VPs. The pair-wise Pearson's correlation of MTAC signals was calculated for each NDR with the rest of 486 NDRs. A 487 × 487 correlation coefficient matrix (Supplementary Data 2) was generated and clustered hierarchically by Wald's method using the scipy.cluster.hierarchy package (version 1.12.0). Heatmap was generated by Seaborn heatmap (version 0.11.1).

## Methylation measurement with bisulfite conversion

For the methylation measurement in Fig. S1B, we generated MTAC strain with LacO inserted in the *MATa* VP and carried out the same MTAC procedure described above. Instead of doing MeDIP-seq, we performed bisulfite conversion and Sanger sequencing to quantify the absolute methylation level of the local region next to the VP in both the MTAC targeted and control strains. 400 ng DNA were extracted after MTAC induction and bisulfite-converted using EZ DNA Methylation-Gold Kit (Zymo Research, D5005). PCR amplification was performed to convert ssDNA to dsDNA using primers designed for a region spanning the *MATa* and the backbone of LacO plasmid used for integration (Supplementary Table 2). PCR product was sequenced by Sanger sequencing. The Methylation rate was calculated by custom MATLAB scripts as the average fraction of "C" at each "GC" position on this PCR product based on the Sanger sequencing intensity of each nucleotide.

## M.CviPI-ID

To identify NPC-associated regions, yeast strains containing *Mlp1-M.CviPI* or free *M.CviPI* (control) was grown in 50 mL synthetic media without methionine at 30 °C to OD660 = 0.3. The expressions of these fusion proteins were induced by adding β-estradiol to a final concentration of 10 nM for 2 h. Methylated DNA was purified and sequenced by MeDIP-seq as previously described. Differentially methylated regions were identified by MACS2 (version 2.2.7.1). NPC-associated NDRs were identified by finding the intersection between NDRs and the differential peaks called by MACS2.

## 3C assay

3C was adapted from a previously published protocol[43]. Yeast strains were grown in 200 mL synthetic media without methionine to OD660 = 0.3. Cells were fixed with 3% formaldehyde (Sigma, 252549) for 20 min at 25 °C and then quenched with glycine for 20 min at room temperature. Cells were collected by centrifugation and washed with the same medium. Cell pellets were resuspended in 1 mL TBS, 1% Triton X-100 and 1X protease inhibitor cocktail (Thermo Fisher, 78420). Cell lysis was performed by adding 500 μL acid-washed glass beads (Sigma, G8772) and vortexing for 25 min at 4 °C. The chromatin was recovered through centrifugation, washed with 1 mL TBS, resuspended in 500 μL 10 mM Tris-HCl buffer and digested with DpnII (NEB, R0543) overnight at 37 °C. The digested DNA fragments were ligated by T4 DNA ligase (Thermo Fisher, EL0014) for 4 h at 16 °C. Crosslink was reversed by incubation with proteinase K at 65 °C overnight. DNA was purified by the standard phenol-chloroform extraction. PCR primers (Supplementary Table 2) were designed for the regions of interest near DpnII cutting sites, and PCR reactions were performed for testing interactions between these regions under different conditions with 4 ng input DNA and 45 amplification cycles.

## 4C assay

4C was adapted from a previously published protocol[65]. Yeast strains were grown in 200 mL synthetic media without methionine to OD660 = 0.3. Cells were fixed with 3% formaldehyde (Sigma, 252549) for 20 min at 25 °C and then quenched with glycine for 20 min at room temperature. Cells were collected by centrifugation and washed with the same medium. Cell pellets were resuspended in 1 mL TBS, 1% Triton X-100 and 1X protease inhibitor cocktail (Thermo Fisher, 78420). Cell lysis was performed by adding 500 μL acid-washed glass beads (Sigma, G8772) and vortexing for 25 min at 4 °C. The chromatin was recovered through centrifugation, washed with 1 mL TBS, resuspended in 500 μL 10 mM Tris-HCl buffer and digested with DpnII (NEB, R0543) overnight at 37 °C. The digested DNA fragments were ligated by T4 DNA ligase (Thermo Fisher, EL0014) for 4 h at 16 °C. Crosslink was reversed by incubation with proteinase K at 65 °C overnight. DNA was purified by the standard phenol-chloroform extraction. A second digestion was performed by resuspending previous 3C DNA template in 1X RE buffer and incubating with NlaIII (NEB, R0125) overnight at 37 °C. The digested DNA fragments were ligated again by T4 DNA ligase overnight at 16 °C. 4C template was purified by phenol-chloroform extraction. VP specific primers were designed to PCR amplify potential interacting DNA fragments of the viewpoint (Supplementary Table 2). 4C-seq libraries were prepared with NEBNext Ultra II DNA Library Prep Kit. Next generation sequencing was performed on NextSeq 2000. Two to five million of 150 bp paired-end reads were aligned to *S. cerevisiae* genome (sacCer3) and analyzed with pipe4C. Peak calling was performed with PeakC.

## Hi-C assay

Hi-C was adapted from a previously published protocol[66]. Yeast strains were grown in 200 mL synthetic media without methionine to OD660 = 0.3. Cells were fixed with 3% formaldehyde for 20 min at 25 °C and then quenched with glycine for 20 min at room temperature. Cells were collected by centrifugation and washed with synthetic media. Cell pellets were resuspended in 1 mL TBS, 1% Triton X-100 and 1X protease inhibitor cocktail. Cell lysis was performed by adding 500 μL acid-washed glass beads and vortexing for 25 min at 4 °C. The chromatin was recovered through centrifugation, washed with 1 mL TBS, resuspended in 500 μL 10 mM Tris-HCl buffer and digested with DpnII overnight at 37 °C. Digested DNA fragments were filled in with biotin-labeled dATP by incubating with Klenow enzyme (NEB, M0212), biotin-14-dATP, dCTP, dTTP, and dGTP for 4 h at room temperature. The biotin-filled DNA fragments were ligated by T4 DNA ligase for 4 h at 16 °C. Crosslink was reversed by incubation with proteinase K at 65 °C overnight. DNA was purified by phenol-chloroform extraction. Biotin-labeled, un-ligated fragment ends were removed by incubating with T4 DNA Polymerase (NEB, M0203), dATP and dGTP for 4 h at 20 °C. DNA was cleaned by DNA clean and concentrator-5 kit (Zymo, D4014) and sheared by Diagenode Bioruptor Pico (EZ mode, 30 s on, 30 s off, 15 cycles). Biotin-labeled DNA was enriched by MyOne™ streptavidin C1 beads (Thermo Fisher, 65001). Hi-C libraries were prepared with NEBNext Ultra II DNA Library Prep Kit. Next generation sequencing was performed on NextSeq 2000. 100 million of 150 bp paired-end reads were aligned to *S. cerevisiae* genome (sacCer3) and analyzed with HiC-pro (version 3.1.0).

## ChIP-seq

Met4 ChIP-seq was performed according to a standard protocol[60]. We tagged the endogenous Met4 with TAP tag and grew the strain in 50 mL synthetic media without methionine at 30 °C to OD660 = 0.3. Cells were crosslinked with 1% formaldehyde, and the reaction was quenched by addition of glycine. Cell pellet was collected and vortexed with glass beads for lysis. Cell lysate was collected again by centrifugation, and sheared by Diagenode Bioruptor Pico (EZ mode, 30 s on, 30 s off, 10 cycles). Sonicated cell lysate was incubated with pre-blocked Magnetic IgG beads and TAP Tag Polyclonal Antibody (Thermo Fisher, CAB1001) for the pull-down of Met4-TAP. Proteinase K was added to reverse the crosslinking. DNA was purified by phenol-chloroform extraction. ChIP and input libraries were prepared with NEBNext Ultra II DNA Library Prep Kit. Next generation sequencing was performed on NextSeq 2000. 5 million of 150 bp paired-end reads were aligned to *S. cerevisiae* genome (sacCer3). Peak calling was performed with MACS2 (version 2.2.7.1).

## RNA-seq

RNA-seq was performed according to a standard protocol[67]. Yeast strains were grown in 50 mL synthetic media with methionine to OD660 = 0.2. The culture is collected by centrifugation and washed 3 x with sterile H₂O. The collected culture is then grown in synthetic media without methionine for 2 h to reach OD660 ~ 0.4 and

centrifuged at 300 g for 4 min at RT. The pellet was washed twice with sterile $H_2O$ and resuspended in 250 μl of RNA lysis buffer (10 mM Tris-HCl, 5 mM EDTA, 2% SDS, 2% stock 2-mercaptoethanol; pH 8.5) and placed on a heat block at 85 °C for 20 min mixing every 2 min. The mixture was centrifuged at 12,000 g for 5 min and the supernatant was transferred to a new tube. The supernatant was then mixed with 1 mL of trizol and heated at 65 °C for 20 min with mixing in between. Standard trizol/chloroform RNA extraction protocol was followed. The extracted RNA was analyzed with TapeStation (Agilent) to check their integrity. RNA with RIN (> 7.5) was used for downstream analysis. mRNA was captured using RNA purification beads (NEB). The eluted mRNA was fragmented and denatured for first and second strand synthesis for conversion into cDNA. The RNA-seq libraries were prepared with NEBNext Ultra II DNA Library Prep Kit. Next generation sequencing was performed on NextSeq 2000. 20 million of 50 bp paired-end reads were aligned to *S. cerevisiae* genome (sacCer3), FPKM was counted by featureCounts (version 2.0.3).

### Mating-type switch assay

The experiment was performed as previously described[43]. Single colonies were grown to log-phase in 50 mL synthetic media with raffinose. *GAL10pr-HO* was induced by adding 3% galactose. After a 2 h incubation, 2% glucose was added to stop the *HO* expression, and cells were incubated for additional 1 h to allow DNA repair. 2 nM of β-estradiol was added during this 3 h process to induce MTAC. Genomic DNA was extracted for genotyping and detection of methylation. To differentiate *MATa* and *MATα*, specific primers were designed to uniquely amplify a or α sequence (Supplementary Table 2). *CDC43* gene was used as control for input genomic DNA.

### Western blotting

Yeast strains were grown in 50 mL synthetic media without methionine to OD660 = 0.5. Cells were split into two samples, one for auxin treatment and one for mock treatment. Auxin was added to the treatment group to a final concentration of 500 μM, and cells were incubated for 1 h. Proteins were extracted by a TCA whole cell extract protocol[68], and western blotting was performed according to standard protocols. Primary antibodies were used at the following concentrations: Anti-V5 tag antibody [SV5-Pk1] (Abcam, ab27671), 1:1000. Anti-Actin antibody [mAbGEa] (Abcam, ab230169), 1:2000. Secondary Anti-Mouse IgG (Fab specific)−Peroxidase antibody produced in goat (Sigma, A9917) was used at 1:2000 concentration.

### Statistics and reproducibility

No statistical method was used to predetermine sample size. No data were excluded from the analyses. For DNA and RNA sequencing experiments, two biological replicates per group were used. For other in vivo experiments, two biological replicates and 3 technical replicates per group were used. The Investigators were not blinded to allocation during experiments and outcome assessment; however, quantifications were performed using computational pipeline and threshold applied equally to all conditions with no bias.

### Reporting summary

Further information on research design is available in the Nature Portfolio Reporting Summary linked to this article.

## Data availability

The sequencing data generated in this study have been deposited into the Gene Expression Omnibus (GEO) database under accession code GSE242400. The viewpoint information and detected chromosomal interactions are listed on Supplementary Data 3. The summary of datasets generated or used in this study is listed on Supplementary Table 3. Source data are provided with this paper.

## Code availability

Custom scripts for correlation analysis and heatmap plot in Fig. 6 are available at github.com/yzl452/MTAC.

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

## Acknowledgements

We thank Dr. Joseph Reese for providing auxin-inducible degron plasmids and yeast strains, and Dr. Michael Kladde for plasmids containing M.CviPI. We acknowledge Dr. Cheryl Keller of the Genomics Research Incubator for technical support. We also want to thank the members of the Center of Eukaryotic Gene Regulation at PSU, especially Dr. Ross Hardison, Dr. Shaun Mahony, Dr. Qunhua Li, and Dr. Joseph Reese, for related discussions. We acknowledge all members in the Bai lab for insightful comments on the manuscript. This work is supported by the National Institutes of Health (T32 GM125592 to Y.L. and R35 GM139654 to L.B.) and the National Science Foundation (MCB- 2016266 to L.B.).

## Author contributions

L.B. and Y.L. designed the experiments; Y.L. performed most of the experiments and data analysis with help from J.L.; Y.L. and L.B. wrote the manuscript.

## Competing interests

The authors declare no competing interest.
