## [Peer Review File · Nature Communications]

DNA Methylation-Based High-Resolution Mapping of Long-Distance Chromosomal Interactions in Nucleosome-Depleted RegionsReviewers' Comments:

Reviewer #1:

Remarks to the Author:

The authors developed a methyltransferase targeting-based assay, named MTAC, to capture high-resolution chromatin contacts between the targeted loci with the rest of the genome in yeast. They further discovered some interesting biological findings in yeast. The approach looks promising compared to 4C. However, I have some major concerns about the technology and data analysis:

1. Off-target effect. I noticed that there is a significant increase in GC methylation at other non-interaction NDR sites. The author utilized the non-LacO array as a control to normalize the background. However, the quantification of off-target events is still not clear in the genome-wide. Although the authors claim that their approach is much better than any other approaches, they should still define a gold standard interacting regions by previous approach (4C, micro-C or others?) in both NDRs and non-NDRs. Then, they need to show how many interacted genome-wide NDRs_vs_NDRs (or NDRs_vs_non-NDRs, non-NDRs_vs_non-NDRs) has been discovered correctly or missed (true positive and false negative rate), and how many of the interactions they discovered are not revealed by other technologies (false positive rate)? Then we will have a clear understanding of their accuracy and power.

2. How many interactions are just due to LacO array insertion? The authors inserted 256X LacO array and use the ones without LacO array insertion as the background. What if a lot of interactions (and GpC methylation) are just due to LacO array insertion rather than nascent chromatin interaction? I mean: their purpose is to detect the nascent interactions and use GpC methylation as the readout. The ideal situation is that LacO array does not change local chromatin and 3D interactions. Then their readout is real to reflect the 3D interactions. What if these read out are just largely due to the artifact of LacO array insertion? The author needs to show that.

3. Why MeDIP but not bisulfite follow-up? They utilized MeDIP instead of BS conversion to measure the methylation level and further interaction. They indeed validate the methylation level and select experiment criteria by using BS conversion. But why they choose to use MeDIP for the final output? We know that BS is a gold standard to measure methylation and MeDIP will suffer for the GpC density bias...

4. Average signals: the author shows a lot of differences genome widely by average plot (Figure S2) or individual loci. To better visualize the differences with the background, a heatmap is needed to better see the differences genome widely. Otherwise, the differences we observed on average (Figure S2) could be just simply due to several limited loci (example plot in main figure)...

5. Validation of far-cis, trans, or weak interacting NDRs. The inflated signals after 100kb in log-log plot (Figure 1C) make me worrisome that a lot of far-cis (>100kb), trans, or weak interacting NDRs are just due to the noise. Since the number is small, the author need to validate them by MeDIP-qPCR.

6. Arbitrary cut-off: not sure why the cut-off is selected as 0.7 for fold-change. Any reason for it?

7. De novo NDRs instead of pre-defined NDRs? Since there is a significant increases of signals at NDR regions even in the input, why not calling NDRs de novo rather than the pre-defined list?

8. NOME-HiC has been developed in human cell lines (GpC methyltransferase combined with Hi-C for the detection of long-range interaction in the same DNA molecules, and validated in region-specific way). As a 4C-like version + GpC methyltransferase, the author failed to mention it in the previous study.

Reviewer #2:

Remarks to the Author:

Li et al. present a novel 3D genomics approach called Methyltransferase Targeting-based chromosome Architecture Capture, MTAC, which is applied to investigate yeast 3D genome organization. The

authors generate a LacI-M. CviPI fusion and induce its expression in vivo. LacI specifically binds the inserted LacO array, targeting the methyltransferase to the locus or viewpoint of interest. Interacting DNA is methylated over a course of 2 hours, generating a stable methylation profile that can be compared to baseline. MTAC eliminates steps in classic 3C-based workflows that introduce biases and contribute to low resolution from cross-linking, digestion, and ligation. The authors demonstrate higher resolution and improved detection of long-range interactions not previously seen in 4C or HiC data. Additionally, this technique can specifically detect NDR interactions absent in previous datasets. MTAC data support expected genomic interactions during mating switch behavior and the authors demonstrate impaired interactions upon depletion of known key factors. Overall, the authors introduce an innovative targeted 3D genomics approach that improves upon existing standard in the field and has the potential to pave the way for new discoveries.

Major Comments

1. It is appreciated that the technique shifts away from restriction enzyme or M_naseI digestion, which introduce sequence bias and digest accessible DNA respectively. The authors instead use M. CviPI to methylate GC dinucleotides. The authors discuss that there are enough GC in accessible regions, "Over 98% of yeast NDR sequences contain "GCs in contrast to 34% containing GATC" (line 77-78). The authors should clarify whether other genomes have enough GC dinucleotides in accessible DNA to support a broad application of this technique beyond yeast.

2. The authors explore MTAC sensitivity to LacO array length and placement, as well as proximity to nuclear pore complexes. They find the number of contacts detected can vary. Have other factors that may sterically impact contact capture been considered? In the placement experiments, were alternative array lengths tested?

Minor Comments

- The conclusion that "tethering to the nuclear pore complex is a major mechanism that mediates long-distance chromosomal interactions in yeast" is overstated and distracts from the important technological advance.
- The authors should clarify the source data of the 4839 NDR regions.
- The authors acknowledge that clustering may change with more viewpoints. This clustering validates that MTAC captures a wide array of interaction types. However the biological relevance of the viewpoint choices is unclear.
- The name of the approach was slightly confusing. As is, MTAC and ATAC-seq describes architecture capture and accessibility capture, respectively.

Reviewer #3:

Remarks to the Author:

Authors present MTAC, a method that maps the contacts between a region of interest (viewpoint, VP) and the interacting regions. MTAC utilizes a GpC methyltransferase, M.CviPI fused to LacI and LacO array probe, which is added to VP. After generating the system, regions interacted are identified by measuring GpC methylation levels. Based on my knowledge, this method is novel. Authors have not only developed and optimized the method but also performed multiple MTAC experiments targeting different regions (VPs) to address some of interesting scientific questions. See below to find my comments and questions.

1. MTAC requires genomic editing before performing the experiments unlike 3C-derivative methods because LacO array needs to be inserted to the VP. In the process of adding it, the original property/activity of the VP can be affected. Moreover, it cannot be applied to tissues or cells which are difficult to be genetically modified. Also, it cannot detect all-to-all genome-wide interactions. The authors need to discuss limitations of MTAC method in discussion section.

2. Include supplemental tables that list experiments performed and datasets generated from this study. It is not clear which experiment is performed in-house and which public datasets were used to generate figures. Also, provide detailed information of each MTAC experiment/data (e.g. location of LacO array/VPs, # of reads sequenced, loop/differentially methylated results, # of reps generated per condition) in supplemental tables, so that other researchers can utilize these data when the manuscript is published.

3. Although I understand that authors have performed many different experiments to optimize the method, but it is not clear how cutoffs are determined. Why did authors use 256X LacO repeat and 2hr? It is not clear why $\log_2(\text{FC}) > 0.7$ and $\text{adjP} < 0.05$ were used. Are these used for all MTAC data analysis or only for Figure 1? Supplemental tables reporting the number of interactions found per different cutoffs and/or different conditions/reps would be beneficial to understand the logic behind the chosen cutoffs. Also, include supplemental tables that show loop/interactions found for each MTAC data.

4. Controls seem to have some signals, and some regions identified as looped regions have quite enriched signals in controls datasets. To ensure that the signals are real, include genome browser screenshots for the same region using additional MTAC data (e.g. MTAC data targeting different VP regions, rep, pseudoreps after shuffling reads to have the same number of reads to the other rep, like IDR (Irreproducibility Discovery Rate) analysis) in supplemental figures.

5. Some interactions are not called from 4C, Micro-C, or Hi-C datasets. It is interesting finding, but not clear if it is due to technical variations (sequencing depth) or background noise (statistical method differences) or biological factors related to experimental differences etc. Include supplemental tables that list which regions are called solely from MTAC or other methods or found common.

6. It is interesting to see that Mlp1 binds to the NDRs interacting each other in the YBP2 locus. When Mlp1 is knocked out, are these interactions change? Are genes interacted with the VP change their expression upon knock out of Mlp1?

7. Figure 6 is too difficult to understand the key points. Perform more focused analyses. For example, perform the analysis after separating loops found intra (Within the chr) and inter (different chromosomes), perform motif search/ChIP-seq overlap analysis to better characterize the loops. What other factors bind to the NDRs not bound by Mlp1? Are loops found trans/far cis related to certain factors? or biased by some MTAC datasets? Technical background/batch effect may affect the trans interaction calls. How many interacting regions/NDRs are found for each MTAC dataset? How many genes are used for each data for Figure 6F? What was the location of VPs and their distances to centromeres (Figure 6F)? Make sure to describe all of MTAC data used to generate the heatmap in supplemental table/figures/methods/GEO in detail.

First, we want to thank all the reviewers for going through our manuscript and providing valuable suggestions that could improve the quality of the paper. The following is a detailed point-by-point reply (blue text) to each comment (black) from the reviewers. We underlined all the sentences describing the changes made to the manuscript. **In the manuscript, the changes have been highlighted in red.**

Reviewer #1 (Remarks to the Author):

The authors developed a methyltransferase targeting-based assay, named MTAC, to capture high-resolution chromatin contacts between the targeted loci with the rest of the genome in yeast. They further discovered some interesting biological findings in yeast. The approach looks promising compared to 4C. However, I have some major concerns about the technology and data analysis:

1. Off-target effect. I noticed that there is a significant increase in GC methylation at other non-interaction NDR sites. The author utilized the non-LacO array as a control to normalize the background. However, the quantification of off-target events is still not clear in the genome-wide. Although the authors claims that their approach is much better than any other approaches, they should still define a gold standard interacting regions by previous approach (4C, micro-C or others?) in both NDRs and non-NDRs. Then, they need to show how many interacted genome-wide NDRs_vs_NDRs (or NDRs_vs_non-NDRs, non-NDRs_vs_non-NDRs) has been discovered correctly or missed (true positive and false negative rate), and how many of the interactions they discovered are not revealed by other technologies (false positive rate)? Then we will have a clear understanding of their accuracy and power.

The off-target effect is indeed important to take into consideration. The increase in GC methylation at other non-interaction NDR sites are the results of activity of free (unbound) M.CviPI that would generate background methylation at genome-wide NDRs. To tackle this problem, we induced M.CviPI at a low level with 2nM β -estradiol (in contrast to 100nM for induction at a saturating level). Nevertheless, some free M.CviPI cannot be avoided. This is why we always compare the targeted vs control strain in our MTAC experiment and use the fold-change of methylation as readout. We found that many viewpoints (like the ones close to NPCs and MAT) only have local interactions, and the number of interactions increase with the number of LacO repeats (Supplementary Fig. 1C). These observations indicate that the genome-wide noise does not generate much, if any, false-positives, and the interactions we detected are largely due to proximity to VP. In case that the point of the control is unclear, we modified a sentence on page 4 to be "The methylation pattern of this yeast strain ("targeted strain") is compared with a control strain with no LacO insertion, which reflects the background methylation by free LacI-M.CviPI."

As far as we know, there are no gold standard interacting regions in budding yeast, as previous 4C, Hi-C, and Micro-C were not able to capture any point-to-point long-distance interactions other than the centromere / telomere clustering. This is different from higher eukaryotes, probably because budding yeast doesn't have distal enhancer and CTCF. However, there are a few evidence supporting the interactions we found in MTAC: 1) the compiled 4C / Hi-C / Micro-C signals over the MTAC far-cis interacting NDRs, as well as compiled 4C signals over the trans interacting NDRs, are significantly higher than the non-interacting regions (Supplementary Fig. 3 E-G). This result indicates that although *individually* these regions are not significant peaks in the 4C / Hi-C / Micro-C datasets, they indeed show higher contact frequency *when pooled together*.

MTAC and 3C-based methods therefore generate consistent results, but MTAC has higher sensitivity and signal-to-noise ratio. 2) MTAC data from the MAT VP show contact with multiple telomeres from different chromosomes, consistent with telomere clustering detected by Hi-C. 3) Some interactions detected here are supported by imaging data, including telomere-MATa interaction¹ and inter-chromosomal clustering of active genes near the NPC². 4) To further verify the interactions detected by MTAC (YBP2 VP), we selected 5 local, 10 far-cis, 10 trans, and 5 negative regions with variable MTAC signal strengths and validated them using low-throughput 3C (see below). Out of the positive regions, 5 local, 8 far-cis, and 8 trans interactions are also detected by 3C, and out of the 5 negative regions, only 1 shows 3C signal.

Importantly, these interactions are independent of LacO-insertion (targeted vs control) and the induction of LacI- M.CviPI (+/- β -estradiol). The 3C data are now included in the manuscript as Supplementary Fig. 4 (referenced on page 6), and 3C methodology is included in the methods (page 18). Overall, these corroborating evidence from Hi-C, imaging, and 3C provide strong support for the MTAC methodology.

2. How many interactions are just due to LacO array insertion? The authors inserted 256X LacO array and use the ones without LacO array insertion as the background. What if a lot of interactions (and GpC methylation) are just due to LacO array insertion rather than nascent chromatin interaction? I mean: their purpose is to detect the nascent interactions and use GpC methylation as the readout. The ideal situation is that LacO array does not change local chromatin and 3D interactions. Then their readout is real to reflect the 3D interactions. What if these read out are just largely due to the artifact of LacO array insertion? The author needs to show that.

We thank the reviewer for raising this concern. As shown in the 3C experiments above, adding the LacO array and/or inducing methylation does not significantly impact the interaction pattern. To further prove this point, we performed Hi-C measurements for the targeted (LacO inserted) and control (non-inserted) strains

with or without induction of methyltransferase (shown on the right in panel A; included as Supplementary Fig. 2). Visually, the Hi-C maps of all datasets are very similar. To analyze this more quantitatively, we generated the correlation between the contact frequencies among each genomic bin (panel B), either throughout the whole genome (left) or over chr VII where the LacO is inserted (right). All correlations are very high (0.94-0.96), comparable to the correlation among biological replica of Hi-C for the same strain under the same condition (typically between 0.9 to 0.96). In a previous study, we also showed that the insertion of 256X LacO repeats, with or without associated LacI, has no impact on cell growth rate³. We therefore conclude that the insertion of 256X LacO has minimal effect on the genome organization and the validated interactions are not artifacts of LacO insertion (added in the text on page 4).

3. Why MeDIP but not bisulfite follow-up? They utilized MeDIP instead of BS conversion to measure the methylation level and further interaction. They indeed validate the methylation level and select experiment criteria by using BS conversion. But why they choose to use MeDIP for the final output? We know that BS is a gold standard to measure methylation and MeDIP will suffer for the GpC density bias...

We indeed tried bisulfite sequencing (BS) in our earlier test of the MTAC method, and it is largely consistent with the MTAC data measured by MeDIP (see the comparison below between MTAC data using MeDIP vs BS, near the MATa VP; MTAC-BS shows the fraction of 5mC in all GC context; zoom-in version of the red box is shown on the right). However, unlike MeDIP, which enriches the methylated DNA, BS sequences through the whole genome regardless of its methylation status, and most of the sequencing data do not carry methylation information (no GC or non-accessible GCs). As a result, we need to sequence a lot more to get sufficient signal-to-noise ratio (typically MeDIP needs 2-5 million reads, while BS needs >50 million). We thus decided to switch to the MeDIP version. We agree with the reviewer that BS provides even higher spatial resolution and suffers less GC bias.

4. Average signals: the author shows a lot of differences genome widely by average plot (Figure S2) or individual loci. To better visualize the differences with the background, a heatmap is needed to better see the differences genome widely. Otherwise, the differences we observed on average (Figure S2) could be just simply due to several limited loci (example plot in main figure)...

We thank the reviewer for this suggestion. We now included the heatmap for interacting and non-interacting NDRs at the YBP2 VP in Supplementary Fig. 3D. All of these interacting loci show significantly higher MTAC signals than the non-interacting ones.

5. Validation of far-cis, trans, or weak interacting NDRs. The inflated signals after 100kb in log-log plot (Figure 1C) make me worrisome that a lot of far-cis (>100kb), trans, or weak interacting NDRs are just due to the noise. Since the number is small, the author need to validate them by MeDIP-qPCR.

All the volcano plots use adjusted P -value < 0.05 as cutoff, i.e. the false-positive rate of identifying interacting peaks should be <0.05. As stated for questions #1, we also validated 25 interactions with 3C and detected 3C signals in 21 of them (included as Supplementary Fig. 4).

6. Arbitrary cut-off: not sure why the cut-off is selected as 0.7 for fold-change. Any reason for it?

0.7 is the cutoff of $\log_2(\text{Fold Change})$. In other words, the methylation level of interacting NDRs need to be >1.62-fold higher in the targeted strain than the control strain. We did not use the more standard 2-fold cutoff because, as stated above, some background methylation cannot be avoided, which limits the range of the fold change. With our typical MTAC sequencing depth (2-5 million reads), we noticed that methylation changes in some NDRs become statistically significant after the $\log_2(\text{Fold Change})$ reaches 0.5, and most MTAC signals with $\log_2(\text{Fold Change}) \geq 0.7$ are statistically significant (see below). We therefore decided to use this number as cut-off throughout the paper.

7. De novo NDRs instead of pre-defined NDRs? Since there is a significant increases of signals at NDR regions even in the input, why not calling NDRs de novo rather than the pre-defined list?

The reviewer is correct in pointing out that the methylation data in principle can be used to map NDRs. However, in this study, we deliberately induce the methyltransferase to a low level to limit the background methylation in genome-wide NDRs. Given the low signal and potential GC bias, we do not think that the methylation data here can generate a very robust NDR map. In contrast, the pre-defined NDRs were derived from MNase data using a sophisticated computational model (Kharerin & Bai, 2021, *PLOS Comp. Bio*)⁴. We applied this model to multiple MNase data and compared our NDR coordinates with the ones annotated from different studies, and all of these results agree to a large extent⁴. We therefore believe that this NDR annotation is more rigorous. We added a sentence in the Method section (page 17) to clarify the source of the NDR coordinates.

8. NOME-HiC has been developed in human cell lines (GpC methyltransferase combined with Hi-C for the detection of long-range interaction in the same DNA molecules, and validated in region-specific way). As a 4C-like version + GpC methyltransferase, the author failed to mention it in the previous study.

This is a very interesting paper, although they mainly use GpC methyltransferase to measure chromatin accessibility, while we are using GpC methyltransferase to capture physical proximity. An important message of this paper is that GpC methyltransferase can be combined with other methods and GpC methylation can be differentiated from CpG methylation through bisulfite sequencing. We referenced this study in the discussion (page 15): “Endogenous 5mC may interfere with MTAC by elevating the background signal in the control and reducing the signal-to-noise ratio. Switching from meDIP to bisulfite sequencing may be beneficial in this case to allow the differentiation of 5mC from different sequence context⁵.”

Reviewer #2 (Remarks to the Author):

Li et al. present a novel 3D genomics approach called Methyltransferase Targeting-based chromosome Architecture Capture, MTAC, which is applied to investigate yeast 3D genome organization. The authors generate a LacI-M. CviPI fusion and induce its expression in vivo. LacI specifically binds the inserted LacO array, targeting the methyltransferase to the locus or viewpoint of interest. Interacting DNA is methylated over a course of 2 hours, generating a stable methylation profile that can be compared to baseline. MTAC eliminates steps in classic 3C-based workflows that introduce biases and contribute to low resolution from cross-linking, digestion, and ligation. The authors demonstrate higher resolution and improved detection of long-range interactions not previously seen in 4C or HiC data. Additionally, this technique can specifically detect NDR interactions absent in previous datasets. MTAC data support expected genomic interactions during mating switch behavior and the authors demonstrate impaired interactions upon depletion of known key factors. Overall, the authors introduce an innovative targeted 3D genomics approach that improves upon existing standard in the field and has the potential to pave the way for new discoveries.

We thank the reviewer for positive comments on our work.

Major Comments

1. It is appreciated that the technique shifts away from restriction enzyme or Mnasel digestion, which introduce sequence bias and digest accessible DNA respectively. The authors instead use M. CviPI to methylate GC dinucleotides. The authors discuss that there are enough GC in accessible regions, “Over 98% of yeast NDR sequences contain “GCs in contrast to 34% containing GATC” (line 77-78). The authors should clarify whether other genomes have enough GC dinucleotides in accessible DNA to support a broad application of this technique beyond yeast.

We thank the reviewer for raising this important point about potential broader applications of MTAC in other species. We added one more paragraph at the end of discussion about potential broader applications of the MTAC (page 15):

“MTAC should have broad applications in other genetically tractable species, including mammals, with some key considerations: 1) GC content in NDRs. Ideally, multiple GCs should be present in individual NDRs to ensure a high signal-to-noise ratio. This criterion should be met in human cells, where NDRs are typically couple of hundred bps in length and GC dinucleotide occurs once every ~20 bp^{6,7}. 2) Stability of the GC methylation. Unlike yeast, many species undergo an active 5mC demethylation process. Although TET enzymes in human cells have a strong preference for 5mCG, they may have lower activities towards 5mC in other sequence context⁸. It thus needs to be carefully evaluated if G5mC is stable enough for MTAC to maintain the interaction information. 3) Native DNA Methylation. Endogenous 5mC may interfere with MTAC by elevating the background signal in the control and reducing the signal-to-noise ratio. Switching from meDIP to bisulfite sequencing may be beneficial in this case to allow the differentiation of 5mC from different sequence context⁵. “

2. The authors explore MTAC sensitivity to LacO array length and placement, as well as proximity to nuclear pore complexes. They find the number of contacts detected can vary. Have other factors that may sterically impact contact capture been considered? In the placement experiments, were alternative array lengths tested?

The variable array lengths were only tested in one locus, and we did not do this with variable distances to the NPCs. If we understand the question correctly, the reviewer is asking whether a long array of LacO present a bulky structure that interferes with the endogenous 3D contacts and/or DNA methylation. We carried out Hi-C measurements in the presence or absence of the inserted LacO array and did not observe significant changes (see more detailed answer for Reviewer #1, question #2). Also, MTAC signal increases with the number of LacO repeats. Therefore, we do not see evidence that long arrays of LacO blocks interaction or methylation.

Minor Comments

- The conclusion that “tethering to the nuclear pore complex is a major mechanism that mediates long-distance chromosomal interactions in yeast” is overstated and distracts from the important technological advance.

The reason we made that statement is because most interactions detected with all the VPs used in this study, except *MAT*, are from NPC-associated NDRs. Out of these VPs, only the ones near *MET* genes have functional considerations, and the other ones are more or less randomly chosen (see below). We therefore think that the interactions we detected should represent typical interactions from random genomic loci in euchromatin (most yeast genome is euchromatin except for a few silenced regions). We agree with the reviewer that stating this in the abstract may be too soon, and we changed the sentence to “most long-distance chromosomal interactions detected by MTAC reflect tethering by the nuclear pore complexes (NPCs)”.

- The authors should clarify the source data of the 4839 NDR regions.

We added a sentence in the Method section (page 17) to clarify the source of the NDR coordinates: “The NDR coordinates were generated by applying a previously described NDR annotation model⁴ to MNase-seq data in Kubik et al. (Supplementary Table 3)⁹.”

- The authors acknowledge that clustering may change with more viewpoints. This clustering validates that MTAC captures a wide array of interaction types. However, the biological relevance of the viewpoint choices is unclear.

The *YBP2* locus was used as a VP because it is located near many NDRs that would provide robust local MTAC signals. All the other VPs in Figure 3 were chosen based on their variable distances to NPCs (in these cases, function of the genes was not part of the consideration). In contrast, VPs near *MET* genes and mating locus were used based on potential genomic interactions that are related to their functions. Out of all the VPs, the only one that was not clearly explained in our original manuscript was *YBP2*, and we thus added the following sentence on page 4: “This VP was chosen because it is located in a gene-rich region near many NDRs that would provide robust local MTAC signals.”

- The name of the approach was slightly confusing. As is, MTAC and ATAC-seq describes architecture capture and accessibility capture, respectively.

We have presented the “MTAC” methods multiple times in conferences, and we did not get the impression that people tend to mix up “MTAC” with “ATAC”. If not absolutely necessary, we would like to keep using this acronym.

Reviewer #3 (Remarks to the Author):

Authors present MTAC, a method that maps the contacts between a region of interest (viewpoint, VP) and the interacting regions. MTAC utilizes a GpC methyltransferase, M.CviPI fused to LacI and LacO array probe, which is added to VP. After generating the system, regions interacted are identified by measuring GpC methylation levels. Based on my knowledge, this method is novel. Authors have not only developed and optimized the method but also performed multiple MTAC experiments targeting different regions (VPs) to address some of interesting scientific questions. See below to find my comments and questions.

1. MTAC requires genomic editing before performing the experiments unlike 3C-derivative methods because LacO array needs to be inserted to the VP. In the process of adding it, the

original property/activity of the VP can be affected. Moreover, it cannot be applied to tissues or cells which are difficult to be genetically modified. Also, it cannot detect all-to-all genome-wide interactions. The authors need to discuss limitations of MTAC method in discussion section.

We thank the reviewer for pointing this out. We added the following sentences to the first paragraph in the discussion (page 13):

“These advantages are accompanied by some drawbacks as well. The MTAC method requires genomic editing with long LacO repeats, which may be challenging for some species and/or cell types. The insertion points of LacO needs to be carefully chosen to avoid perturbation of cellular function. Also, it is hard for MTAC to be scaled up to detect all-to-all interactions like Hi-C.”

2. Include supplemental tables that list experiments performed and datasets generated from this study. It is not clear which experiment is performed in-house and which public datasets were used to generate figures. Also, provide detailed information of each MTAC experiment/data (e.g. location of LacO array/VPs, # of reads sequenced, loop/differentially methylated results, # of reps generated per condition) in supplemental tables, so that other researchers can utilize these data when the manuscript is published.

We included the VP information and differential methylation results (interactions) in Supplementary Table 5. We summarized all the experiments and datasets in Supplementary Table 6, including the source, data type, # of reads, and # of repeats. They can also be accessed on GEO (GSE242400). We added the following sentence to the “data availability” section: “All detected interactions and viewpoint information are listed on Supplementary Table 5. The summary of datasets used in this study is listed on Supplementary Table 6.”

3. Although I understand that authors have performed many different experiments to optimize the method, but it is not clear how cutoffs are determined. Why did authors use 256X LacO repeat and 2hr? It is not clear why $\log_2(\text{FC}) > 0.7$ and $\text{adjP} < 0.05$ were used. Are these used for all MTAC data analysis or only for Figure 1? Supplemental tables reporting the number of interactions found per different cutoffs and/or different conditions/reps would be beneficial to understand the logic behind the chosen cutoffs. Also, include supplemental tables that show loop/interactions found for each MTAC data.

Please see our answer to reviewer #1, question #6, for why we chose 0.7 as the cutoff for \$\log_2(\text{FC})\$. Using 0.05 for \$\text{Padj}\$ is standard in the field (same cutoff for the 3C based datasets, also uses \$\text{FDR} < 0.05\$ ). The same cutoffs are used for all MTAC data (added note on page 5). All detected interactions and viewpoint information are listed on Supplementary Table 5.

4. Controls seem to have some signals, and some regions identified as looped regions have quite enriched signals in controls datasets. To ensure that the signals are real, include genome browser screenshots for the same region using additional MTAC data (e.g. MTAC data targeting different VP regions, rep, pseudoreps after shuffling reads to have the same number of reads to the other rep, like IDR (Irreproducibility Discovery Rate) analysis) in supplemental figures.

For the background methylation signals in the control strain, please refer to our answer to Reviewer #1, question 1. We generated pseudoreps by combining, shuffling, and splitting reads

for the YBP2 VP. With the pseudoreps, we detected more interactions using the same threshold as a result of reduced variation between replicates (See figure below). It's noted that previously detected interactions using the true replicates are a subset of newly identified interactions using the pseudoreps, indicating that these are interactions detected with high confidence. We show some examples below with data from the true replicates of YBP2 VP, pseudo-replicates, *MLP1/MLP2* knock-out mutant (See more discussion below for question 6), and data from a different VP *MATa*. These regions are positive in the true and pseudo replicates of YBP2 VP, but negative in *MLP1/MLP2* mutant and *MATa* VP. Some of the comparisons are added to Fig 2. We also validated 21/25 interactions with 3C in Supplementary Fig. 4B.

5. Some interactions are not called from 4C, Micro-C, or Hi-C datasets. It is interesting finding, but not clear if it is due to technical variations (sequencing depth) or background noise (statistical

method differences) or biological factors related to experimental differences etc. Include supplemental tables that list which regions are called solely from MTAC or other methods or found common.

There is evidence that our MTAC measurement is consistent with 3C-based or imaging-based methods (see our answer to Reviewer #1, Question #1). However, it is also important to note that MTAC and 3C methods have some fundamental differences, making it hard to directly compare their signal strength. The main difference is that 3C method takes “*snapshots*” using fixed cells while MTAC signal is *accumulative over time* in live cells. Many long-distance interactions are highly dynamic (based on imaging data, almost no long-distance contacts can simultaneously show up in more than 10% of cells), and therefore their contact signals are largely diluted in a population of cells. In contrast, MTAC uses a semi-stable methylation signal to record interactions over time so that all cell population that has made contacts during the 2hr methyltransferase induction will carry the interaction signal (not just the ones that are making contacts now). Therefore, MTAC is more sensitive to long-distance interactions.

A fair comparison can be made between MTAC and 4C, which measures interactions at the same scale. With similar sequencing depth (both 2-5 million reads) and similar statistical cutoff (FDR < 0.05), MTAC from some VPs can detect a lot more long-distance interactions than 4C. With YBP2 as the viewpoint, for example, MTAC detects 22 far-cis interactions, while only one significant intra-chromosomal interactions can be called using 4C.

6. It is interesting to see that Mlp1 binds to the NDRs interacting each other in the YBP2 locus. When Mlp1 is knocked out, are these interactions change? Are genes interacted with the VP change their expression upon knock out of Mlp1?

We thank the reviewer for this insightful question. Mlp1 has a homolog, Mlp2, which is functionally redundant. To test this idea, we constructed double knock-out of *MLP1* and *MLP2* and performed MTAC on the same *YBP2* VP. We found that the *MLP1/MLP2* are indeed essential for the long-distance interactions of *YBP2* (see panel E-H in the figure below). We added the data of *MLP1/MLP2* mutant to Fig 2, and described the finding in the manuscript on page 7:

“The results above strongly indicate the critical role of NPC in mediating long-distance genomic interactions with *YBP2* VP. To further test this idea, we knocked out *MLP1* and its paralog *MLP2* and performed MTAC on the same *YBP2* VP. In this *MLP1/MLP2* double-knockout mutant, the local interactions of *YBP2* remain largely intact (**Fig 2E & F**), but most of the far-*cis* interactions and all the *trans* interactions are lost (**Fig 2E & G**). Quantitatively, MTAC signals over long-distance interacting NDRs are significantly decreased, but the ones over non-interacting NDRs are slightly increased in the mutant (**Fig 2H**), indicating a global deregulation of genome organization. Together, these results show that the contacts with *YBP2* VP from distant genomic loci are likely due to the tethering by NPCs, and Mlp1 / Mlp2 are essential to mediate these interactions (**Fig 2I**).”

We did not measure the change of gene expression in the *MLP1/MLP2* double-knockout strain as it is beyond the scope of the current manuscript.

7. Figure 6 is too difficult to understand the key points. Perform more focused analyses. For example, perform the analysis after separating loops found intra (Within the chr) and inter (different chromosomes), perform motif search/ChIP-seq overlap analysis to better characterize the loops. What other factors bind to the NDRs not bound by Mlp1? Are loops found trans/far cis related to certain factors? or biased by some MTAC datasets? Technical background/batch effect may affect the trans interaction calls. How many interacting regions/NDRs are found for each MTAC dataset? How many genes are used for each data for Figure 6F? What was the location of VPs and their distances to centromeres (Figure 6F)? Make sure to describe all of MTAC data used to generate the heatmap in supplemental table/figures/methods/GEO in detail.

First, we want to clarify the point of Figure 6. All data in Figure 1-5 address the question that, for each selected VP, which NDRs it contacts. After collecting interaction data for 15 VPs, Figure 6 asks a reciprocal question, i.e. for each NDR, which VPs it contacts. In other words, no new data were generated for Figure 6. We reanalyzed the data in Figure 1-5 in a different way.

For NDRs with a very similar interaction pattern, it is likely that these NDRs occupy similar nuclear space, and vice versa. For example, in a hypothetical case listed below where “√” and “x” indicate contact and no contact, respectively, we would conclude that NDR1 and 2 are close to each other, and so are 3 and 4, but 1 & 2 are physically separated from 3 & 4.

	VP1	VP2	VP3	VP4	VP5	VP6	VP7	VP8	VP9	VP10
NDR1	×	×	✓	✓	✓	×	✓	✓	✓	×
NDR2	×	×	✓	✓	×	×	✓	✓	✓	×
NDR3	✓	✓	×	×	×	✓	×	×	×	✓
NDR4	✓	✓	×	×	×	✓	×	✓	×	✓

We clarified this point in the Discussion (page 14). The number of NDRs that belong to each cluster, as well as the number of genes used for Figure 6F, are now included in the figure legend. As mentioned above for question #2, all raw and analyzed MTAC data are listed in Supplementary Table 5, Supplementary Table 6, and GEO (GSE242400). All raw data for generating quantitative plots are provided in Source Data.

The reviewer also asked an interesting question if there are other factors besides Mlp1 that may explain the interaction pattern. To address this question, we performed bioinformatic analysis to find factors that are enriched on the Cluster 1 NDRs using ChIP-exo data of 105 TFs. Mlp1 is indeed one of the most enriched factors that are associated with these NDRs (see plot below). A few other enriched factors are also found, especially Rsc3, a subunit of RSC chromatin remodeler. Interestingly, RSC was found to physically interact with NPC and regulate its localization^{10,11}. Another enriched TF, Sfl1, was also shown to mediate translocation of its target genes to NPC². Therefore, the enrichment of these TFs may reflect their interactions with NPC, although the exact mechanism requires further investigation. No significant TFs were found on NDRs in other clusters, probably due to low statistics. Because of the preliminary nature of these data, we decided not to include them in the current manuscript, but we can if the reviewer thinks that this is beneficial for the overall quality of the paper.

References

- 1 Belton, J. M. *et al.* The Conformation of Yeast Chromosome III Is Mating Type Dependent and Controlled by the Recombination Enhancer. *Cell Rep* **13**, 1855-1867 (2015). <https://doi.org/10.1016/j.celrep.2015.10.063>
- 2 Brickner, D. G. *et al.* The Role of Transcription Factors and Nuclear Pore Proteins in Controlling the Spatial Organization of the Yeast Genome. *Dev Cell* **49**, 936-947.e934 (2019). <https://doi.org/10.1016/j.devcel.2019.05.023>
- 3 Du, M., Zou, F., Li, Y., Yan, Y. & Bai, L. Chemically Induced Chromosomal Interaction (CICI) method to study chromosome dynamics and its biological roles. *Nat Commun* **13**, 757 (2022). <https://doi.org/10.1038/s41467-022-28416-3>
- 4 Kharerin, H. & Bai, L. Thermodynamic modeling of genome-wide nucleosome depleted regions in yeast. *PLoS Comput Biol* **17**, e1008560 (2021). <https://doi.org/10.1371/journal.pcbi.1008560>
- 5 Fu, H. *et al.* NOMe-HiC: joint profiling of genetic variant, DNA methylation, chromatin accessibility, and 3D genome in the same DNA molecule. *Genome Biol* **24**, 50 (2023). <https://doi.org/10.1186/s13059-023-02889-x>
- 6 Oszolak, F., Song, J. S., Liu, X. S. & Fisher, D. E. High-throughput mapping of the chromatin structure of human promoters. *Nat Biotechnol* **25**, 244-248 (2007). <https://doi.org/10.1038/nbt1279>
- 7 Nakashima, H., Nishikawa, K. & Ooi, T. Differences in dinucleotide frequencies of human, yeast, and Escherichia coli genes. *DNA Res* **4**, 185-192 (1997). <https://doi.org/10.1093/dnares/4.3.185>
- 8 Ravichandran, M. *et al.* Pronounced sequence specificity of the TET enzyme catalytic domain guides its cellular function. *Sci Adv* **8**, eabm2427 (2022). <https://doi.org/10.1126/sciadv.abm2427>
- 9 Kubik, S. *et al.* Opposing chromatin remodelers control transcription initiation frequency and start site selection. *Nat Struct Mol Biol* **26**, 744-754 (2019). <https://doi.org/10.1038/s41594-019-0273-3>
- 10 Van de Vosse, D. W. *et al.* A role for the nucleoporin Nup170p in chromatin structure and gene silencing. *Cell* **152**, 969-983 (2013). <https://doi.org/10.1016/j.cell.2013.01.049>
- 11 Titus, L. C., Dawson, T. R., Rexer, D. J., Ryan, K. J. & Wentz, S. R. Members of the RSC chromatin-remodeling complex are required for maintaining proper nuclear envelope structure and pore complex localization. *Mol Biol Cell* **21**, 1072-1087 (2010). <https://doi.org/10.1091/mbc.e09-07-0615>

Reviewers' Comments:

Reviewer #1:

Remarks to the Author:

The reviewer has addressed all my concerns.

Reviewer #2:

Remarks to the Author:

The authors have sufficiently addressed both major and minor concerns raised. With these updates, I recommend this paper for publication.

Reviewer #3:

Remarks to the Author:

Authors addressed most of the questions I asked. However, it is not clear why double knock-out of Mlp1 and Mlp2 experiment was performed instead of the single knock-out of Mlp1 experiment. Is single knock-out Mlp1 MTAC data generated but no interaction difference is observed due to Mlp2? Did Mlp2 expression go up in the single knock-out of Mlp1 cells? Is Mlp2 also found as one of the most enriched factors? I do not see that Mlp2 is highlighted in the scatterplot of the rebuttal Q7. It is interesting to see that "the local interactions of YBP2 remain largely intact, but most of the far-cis interactions and all the trans interactions are lost in the double knock-out". To confirm that it is not due to technical issues or variations, include Mlp1 and Mlp2 KO Western Blot result and MTAC signal plots (heatmaps, genome browser screenshots) at other genomic loci.

We are pleased that we have completely addressed the concerns of the first two reviewers, and they have approved our manuscript. The following is a reply to the third reviewer (blue text). The corresponding changes in the manuscript are highlighted in red.

Reviewer #3 (Remarks to the Author):

Authors addressed most of the questions I asked. However, it is not clear why double knock-out of Mlp1 and Mlp2 experiment was performed instead of the single knock-out of Mlp1 experiment. Is single knock-out Mlp1 MTAC data generated but no interaction difference is observed due to Mlp2? Did Mlp2 expression go up in the single knock-out of Mlp1 cells? Is Mlp2 also found as one of the most enriched factors? I do not see that Mlp2 is highlighted in the scatterplot of the rebuttal Q7. It is interesting to see that "the local interactions of YBP2 remain largely intact, but most of the far-cis interactions and all the trans interactions are lost in the double knock-out". To confirm that it is not due to technical issues or variations, include Mlp1 and Mlp2 KO Western Blot result and MTAC signal plots (heatmaps, genome browser screenshots) at other genomic loci.

It has been shown that many nuclear pore proteins have overlapping functions. Nuclear basket protein Mlp1 and Mlp2 are paralogs that arose from the whole genome duplication of yeast. They both localize to the periphery of NPC¹ and interact with similar proteins². It's been shown that single knock-out of Mlp1 or Mlp2 has no effect on yeast growth in rich medium, while only double knock-out shows growth defect compared to the WT¹. These studies suggest that Mlp1 and Mlp2 are largely redundant in their functions and likely compensate each other when one is depleted. Past studies of nuclear baskets always simultaneously deleted both proteins¹⁻⁶. We therefore followed this common practice in the field. We changed the sentence on page 7 to clarify this point: "To further test this idea, we deleted *MLP1* and its paralog *MLP2*, as some studies have suggested that these two proteins have largely redundant functions.^{1,2"} We did not perform MTAC on Mlp1 single knock-out strain.

Mlp2 does not appear on the scatterplot for Q7 because there is no high-throughput sequencing data available for Mlp2. Given the co-localization of Mlp1 and Mlp2, we expect the two proteins to contact similar genomic regions. We deleted Mlp1 and Mlp2 by replacing *MLP1* and *MLP2* genes with KanMX and HygroMX markers, respectively. We are confident that these deletions are carried out properly because we not only did PCR-based genotyping after the transformation, but

also no reads can be mapped to the *MLP1/MLP2* gene in our MTAC sequencing data, indicating complete elimination of these sequences (shown below).

As requested by the reviewer, we generated the heatmap of the MTAC data in the *mlp1/mlp2* double deletion strain, as well as more examples of individual regions (shown below with side-by-side comparison to WT). We included these data in the new Supplementary Fig S6.

References

- 1 Strambio-de-Castillia, C., Blobel, G. & Rout, M. P. Proteins connecting the nuclear pore complex with the nuclear interior. *J Cell Biol* **144**, 839-855 (1999). <https://doi.org/10.1083/jcb.144.5.839>
- 2 Niepel, M. *et al.* The nuclear basket proteins Mlp1p and Mlp2p are part of a dynamic interactome including Esc1p and the proteasome. *Mol Biol Cell* **24**, 3920-3938 (2013). <https://doi.org/10.1091/mbc.E13-07-0412>
- 3 Galy, V. *et al.* Nuclear pore complexes in the organization of silent telomeric chromatin. *Nature* **403**, 108-112 (2000). <https://doi.org/10.1038/47528>

- 4 Zhao, X., Wu, C. Y. & Blobel, G. Mlp-dependent anchorage and stabilization of a desumoylating enzyme is required to prevent clonal lethality. *J Cell Biol* **167**, 605-611 (2004). <https://doi.org/10.1083/jcb.200405168>
- 5 García-Benítez, F., Gaillard, H. & Aguilera, A. Physical proximity of chromatin to nuclear pores prevents harmful R loop accumulation contributing to maintain genome stability. *Proc Natl Acad Sci U S A* **114**, 10942-10947 (2017). <https://doi.org/10.1073/pnas.1707845114>
- 6 Zsok, J. *et al.* The nuclear basket regulates the distribution and mobility of nuclear pore complexes in budding yeast. *bioRxiv*, 2023.2009. 2028.558499 (2023).

Reviewers' Comments:

Reviewer #3:

Remarks to the Author:

The reviewer has addressed my concerns.